# Safety through feedback in Constrained RL

**Shashank Reddy Chirra**[*1], **Pradeep Varakantham**[1], **Praveen Paruchuri**[2]
[1]Singapore Management University, [2]IIIT Hyderabad
{shashankc,pradeepv}@smu.edu.sg, praveen.p@iiit.ac.in

## Abstract

In safety-critical RL settings, the inclusion of an additional cost function is often favoured over the arduous task of modifying the reward function to ensure the agent's safe behaviour. However, designing or evaluating such a cost function can be prohibitively expensive. For instance, in the domain of self-driving, designing a cost function that encompasses all unsafe behaviours (e.g., aggressive lane changes, risky overtakes) is inherently complex, it must also consider all the actors present in the scene making it expensive to evaluate. In such scenarios, the cost function can be learned from feedback collected offline in between training rounds. This feedback can be system generated or elicited from a human observing the training process. Previous approaches have not been able to scale to complex environments and are constrained to receiving feedback at the state level which can be expensive to collect. To this end, we introduce an approach that scales to more complex domains and extends beyond state-level feedback, thus, reducing the burden on the evaluator. Inferring the cost function in such settings poses challenges, particularly in assigning credit to individual states based on trajectory-level feedback. To address this, we propose a surrogate objective that transforms the problem into a state-level supervised classification task with noisy labels, which can be solved efficiently. Additionally, it is often infeasible to collect feedback for every trajectory generated by the agent, hence, two fundamental questions arise: (1) Which trajectories should be presented to the human? and (2) How many trajectories are necessary for effective learning? To address these questions, we introduce a *novelty-based sampling* mechanism that selectively involves the evaluator only when the the agent encounters a *novel* trajectory, and discontinues querying once the trajectories are no longer *novel*. We showcase the efficiency of our method through experimentation on several benchmark Safety Gymnasium environments and realistic self-driving scenarios. Our method demonstrates near-optimal performance, comparable to when the cost function is known, by relying solely on trajectory-level feedback across multiple domains. This highlights both the effectiveness and scalability of our approach. The code to replicate these results can be found at https://github.com/shshnkreddy/RLSF

## 1 Introduction

Reinforcement Learning (RL) is known to suffer from the problem of reward design, especially in safety-related settings [4]. In such a case, constrained RL settings have emerged as a promising alternative to generate safe policies[3, 16, 34]. This framework introduces an additional cost function to split the task related information (rewards) from the safety related information (costs). In this paper, we address a scenario where the reward function is well-defined, while the cost function remains unknown *a priori* and requires inference from feedback. This arises in cases where the cost function is *expensive* to design or evaluate. For instance, consider the development of an autonomous

---

[*]Corresponding Author

38th Conference on Neural Information Processing Systems (NeurIPS 2024).

driving system, where the reward function may be defined as the time taken to reach a destination which is easy to define. However, formulating the cost function presents significant challenges. Designing a comprehensive cost function that effectively penalizes all potential unsafe behaviors is non-trivial, and ensuring safety often involves subjective judgments, which can vary based on individual preferences. Even if one succeeds in devising such a function, it must account for all environmental factors, such as neighboring vehicles and pedestrians. The evaluation of such a cost function in high-fidelity simulators can be prohibitively expensive.

Feedback can be collected from either a human observer, who monitors the agent's training process and periodically provides feedback on presented trajectories, or from a system that computes the cost incurred on selected trajectories. Throughout this paper, we use the term *evaluator* to refer to the entity providing feedback, whether human or system-generated.

Cost inference in Constrained RL settings has gained recent attention. One key thread of research has focused on learning cost from from constraint abiding expert demonstrations [28, 32]. However, these expert demonstrations are not easily available in all settings. For example, consider robotic manipulation tasks where the human and the robot have different morphologies. This paper focuses on the second line of research in this area, where we generate trajectories and gather feedback from an evaluator to infer the underlying cost function. Prior works in this thread make limiting assumptions on the nature of the cost function such as *smoothness* or *linearity* [28, 32, 11, 5] and are limited to obtaining feedback at the state level [8] which is expensive to collect from human evaluators. We do not make such assumptions and can take feedback provided at over longer horizons, in some cases the entire trajectory.

Frameworks for learning from feedback must exhibit the following properties as emphasized in [8, 18, 12]: 1) Feedback must be collected *offline* in between rounds, since the agent may need to act in real time. 2) The amount of feedback collected must be *minimized*. 3) *Binary* feedback is more suitable as compared to *numeric* feedback as it is more intuitive to provide for humans. It has also been shown that humans provide less consistent feedback if it is *numeric* [12]. 4) Each state is assigned a binary cost value, indicating that it is inherently *safe* or *unsafe*, which is more intuitive for humans when assessing the safety of policies [31] [1].

To this end we propose the **R**einforcement **L**earning from **S**afety **F**eedback (**RLSF**) algorithm, which embodies all the aforementioned properties. The key contributions of this algorithm include:

- Extends prior work to collect feedback over longer horizons. This is done by presenting the evaluator with the entire trajectory, breaking the trajectory into segments and eliciting feedback at the segment level. Inferring the costs directly by *maximizing likelihood* presents new challenges due to the problem of credit assignment over longer horizons. To tackle this, we present a surrogate loss that converts the problem from trajectory level cost inference to a supervised binary classification problem with *noisy* labels.

- Introduces a *novelty based sampling* mechanism that reduces the number of queries by sampling *novel* trajectories for feedback.

- Learns safe policies across diverse benchmark safety environments. We also show that the learnt cost function can be *transferred* to train an agent with different dynamics/morphology from scratch without collecting additional feedback.

## 2   Preliminaries

**Markov Decision Process**   A Markov Decision Process (MDP) $\mathcal{M}$ is defined by the tuple $(\mathcal{S}, \mathcal{A}, \mathcal{P}, r, \gamma, \mu)$, where $\mathcal{S}$ denotes the set of states, $\mathcal{A}$ is the set of actions, $\mathcal{P}(s'|s, a) \in [0, 1]$ is the transition probability, $r(s, a) \in \mathbb{R}$ is the reward function, $\gamma \in [0, 1]$ is the discount factor and $\mu(s) \in \Delta(\mathcal{S})$ is the initial state distribution. A policy $\pi(.|s) \in \Delta(\mathcal{A})$ is a distribution over the set of valid actions for state $s$. We denote the set of all stationary policies as $\Pi$. A *trajectory* $\tau = \{(s_t, a_t)\}$ denotes the state-action pairs encountered by executing $\pi$ in $\mathcal{M}$. We use the short hand $\tau_{i:j}$ to denote a *trajectory segment*, i.e, the subsequence of $(s_t, a_t)$ pairs encountered from timestep $i$ to $j$. The *expected value* of a function $f$ under $\pi$ as $\mathcal{J}^f(\pi) \triangleq E_{\tau \sim \pi}[\sum_{t=0}^{\infty} \gamma^t f(s_t, a_t)]$. We also employ the shorthand $f(\tau)$ to represent the discounted sum of $f$ along the trajectory $\tau$. The occupancy measure

---

[1]The last two points are specific to human evaluators

of a policy is define as $\rho(s,a) = E_{\tau \sim \pi}[\sum_{t=0}^{\infty} \gamma^t \mathbb{I}[(s_t, a_t) = (s,a)]]$, where $\mathbb{I}[.]$ denotes the indicator function. $\rho$ describes the frequency with which a state-action pair is visited by $\pi$.

**Constrained Markov Decision Process** A Constrained MDP [4] introduces a function $c(s,a) \in \mathbb{R}$ and a cost threshold $c_{max} \in \mathbb{R}$ that defines the maximum cost that can be accrued by a policy. The set of feasible policies is defined as $\Pi_c = \{\pi \in \Pi : \mathcal{J}^c(\pi) \leq c_{max}\}$. A policy is considered to be *safe* w.r.t $c$ if it belongs to $\Pi_c$.

## 3 Problem Definition

In this paper, we consider the constrained RL problem defined as,

$$\pi^* = \underset{\pi \in \Pi_c}{\operatorname{argmin}} \ \mathcal{J}^r(\pi) \tag{1}$$

We assume the threshold $c_{max}$ is known, but the cost function is not known and must be inferred from feedback collected from an external evaluator. In many scenarios $c_{max}$ is typically known, representing a predefined limit on acceptable costs or risks in the environment. However, crafting the cost function $c(s,a)$ such that it penalizes all *unsafe* behaviour can be infeasible.

We incorporate an additional constraint enforcing the cost function to be binary, i.e, $c(s,a) \in \{0,1\}$. This ensures that each state-action pair is inherently categorized as either *safe* or *unsafe*. We opt for this approach because it is simpler for human evaluators to assign a binary safety value to state-actions when assessing policy safety, as emphasized in [31].

## 4 Method

In this section, we introduce **R**einforcement **L**earning from **S**afety **F**eedback (**RLSF**), an on-policy algorithm that consists of two alternating stages: 1) Data/Feedback collection and 2) Constraint inference/Policy improvement. In the first stage, data is collected via rollouts of the current policy for a fixed number of trajectories. Next, a subset of these trajectories is presented for feedback from evaluator, which are then stored in a separate buffer. The second stage consists of two parts: i) Estimation of the cost function from the feedback data and ii) Improvement of the policy using the collected trajectories and their inferred costs. We repeat stages (1) and (2) until convergence.

First, we highlight how the feedback is collected and propose a method to infer the constraint function using this data. Next, we recognize the practical limitations of acquiring feedback for every trajectory during training and detail our approach to sampling a subset of trajectories for efficient cost learning of the cost function. Finally, we detail how the inferred cost function is used to improve the policy.

### 4.1 Nature of the Feedback

In the feedback process, the evaluator is first presented with the entire trajectory $\tau_{0:T}$. Afterward, the trajectory is divided into contiguous segments $\tau_{i:j}$ of length $k$, and feedback is collected for each segment. The segment length can be adjusted based on the complexity of the environment: in simpler environments, feedback can be gathered for the entire trajectory, whereas for environments with long horizons and sparse cost violations, shorter segments may be used. This approach simplifies the challenge of assigning credit to individual states. Similar methods have been adopted in other works that rely on human feedback [12, 18]. However, reducing segment length comes at an increased cost of obtaining feedback from the evaluator.

The evaluator is tasked with classifying a segment as *unsafe* if the agent encounters an *unsafe* state at any point within the segment. This decision was made to ensure consistent feedback from the evaluator. Alternative approaches—such as marking a segment *unsafe* based on the number of unsafe states visited or leaving the classification to the evaluator's subjective judgment are more prone to generating inconsistent feedback in the case of human evaluators [12].

### 4.2 Inferring the Cost Function

Let $P = \{\tau_{i:j}, y^{safe}\}$ represent the feedback collected from the evaluator, where $y^{safe} = 1$ if the segment was labelled *safe* and $y^{safe} = 0$ otherwise. We assume there exists an underlying ground

truth cost function $c_{gt}(s, a) \in [0, 1]$ based on which the evaluator provides feedback. Then, the probability that a state is *safe* is defined as $p_{gt}^{safe}(s, a) = \mathbb{I}[c_{gt}(s, a) = 0]$.

Now, let $p^{safe}(s, a)$ represent our estimate of $p_{gt}^{safe}(s, a)$ that we intend to estimate from the collected feedback. Then, by definition of how the feedback is collected, the probability that a trajectory segment $\tau_{i:j}$ is labelled as *safe* is given by,

$$p^{safe}(\tau_{i:j}) = \prod_{t=i}^{j} p^{safe}(s_t, a_t) \tag{2}$$

We can infer $p^{safe}(s, a)$ by minimizing the likelihood loss,

$$L^{mle} = -E_{(\tau_{i:j}, y^{safe}) \sim P} \left[ y^{safe} \log p^{safe}(\tau_{i:j}) + (1 - y^{safe}) \log 1 - p^{safe}(\tau_{i:j}) \right]$$

$$= -E_{(\tau_{i:j}, y^{safe}) \sim P} \left[ y^{safe} \sum_{t=i}^{j} \log p^{safe}(s_t, a_t) + (1 - y^{safe}) \log \left(1 - \prod_{t=i}^{j} p^{safe}(s_t, a_t)\right) \right] \tag{3}$$

Directly minimizing Eq 3 is challenging as the term $\prod_{t=i}^{j} p^{safe}(s_t, a_t)$ would collapse to 0 when the segment length is long, causing unstable gradients. To address this issue, we propose using a surrogate loss function where we replace $1 - \prod_{t=i}^{j} p^{safe}(s_t, a_t)$ by $\prod_{t=i}^{j} (1 - p^{safe}(s_t, a_t))$.

$$L^{sur} = -E_{(\tau_{i:j}, y^{safe}) \sim P} \left[ y^{safe} \sum_{t=i}^{j} \log p^{safe}(s_t, a_t) + (1 - y^{safe}) \sum_{t=i}^{j} \log \left(1 - p^{safe}(s_t, a_t)\right) \right]$$

$$= -E_{(\tau_{i:j}, y^{safe}) \sim P} \sum_{t=i}^{j} \sum_{s,a} \mathbb{I}[(s_t, a_t) = (s, a)] \left[ \mathbb{I}[y^{safe} = 1] \log p^{safe}(s, a) \right.$$

$$\left. + \mathbb{I}[y^{safe} = 0] \log \left(1 - p^{safe}(s, a)\right) \right]$$

$$= -\left[ E_{(s,a) \sim d_g} \log p^{safe}(s, a) + E_{(s,a) \sim d_b} \log \left(1 - p^{safe}(s, a)\right) \right] \tag{4}$$

where $d_g(s, a) = E_{(\tau_{i:j}, y_{safe}) \sim P} \left[ \sum_{t=i}^{j} \mathbb{I}[(s_t, a_t) = (s, a) \cap y^{safe} = 1] \right]$ and $d_b(s, a) = E_{(\tau_{i:j}, y_{safe}) \sim P} \left[ \sum_{t=i}^{j} \mathbb{I}[(s_t, a_t) = (s, a) \cap y^{safe} = 0] \right]$ represent the densities with which states occur in *safe* and *unsafe* segments respectively.

The surrogate loss reformulates the objective from the segment level—where collapsing probabilities over long segments can be problematic—to the state level, where this issue does not occur. Optimizing Eq 4 involves breaking down the segments into individual states and assigning each state the label of the segment. Subsequently, states are uniformly sampled at random, and the binary cross-entropy loss is minimized. Consequently, $L^{sur}$ represents a binary classification problem in which one class contains *noisy* labels. While *unsafe* states are always accurately labeled—since their presence necessitates an *unsafe* classification for the segment—a *safe* state may receive conflicting labels based on the status of the segment it belongs to. Thus, with a sufficient number of samples, we believe it becomes feasible to reliably differentiate between *safe* and *unsafe* states.

**Proposition 1.** *The surrogate loss $L^{sur}$ is an upper bound on the likelihood loss $L^{mle}$.*

Thus, minimizing $L^{sur}$ guarantees an upper bound on the likelihood loss of the estimated cost function.

Having discussed the surrogate loss, we now examine the characteristics of its optimal solution.

**Proposition 2.** *The optimal solution to Eq 4 yields the estimate,*

$$p_*^{safe}(s, a) = \frac{d_g(s, a)}{d_g(s, a) + d_b(s, a)} \tag{5}$$

Subsequently, we define the inferred cost function as $c_*(s, a) \triangleq \mathbb{I}[p_*^{safe}(s, a) < \frac{1}{2}]$. Employing $c_*(s, a)$ in policy updates instead of $c_{gt}(s, a)$ introduces a bias in estimating the cost accrued by the

policy, that we analyse below. For this analysis, we assume that the feedback is *sufficient*, i.e, the density $d(s, a)$ is greater than zero for every state, otherwise $p^*(s, a)$ is not defined.

**Proposition 3.** *For a fixed policy $\pi$, the bias in the estimation of the incurred costs is given by,*

$$E_\pi[\gamma^t c_*(s, a)] - E_\pi[\gamma^t c_{gt}(s, a)] = E_{(s,a)\sim\rho_g^\pi}[\mathbb{I}[d_b(s, a) > d_g(s, a)]] \tag{6}$$

*where $\rho_g^\pi(s, a) = E_\pi[\sum_{t=0}^T \gamma^t[\mathbb{I}[(s_t, a_t) = (s, a) \cap c_{gt}(s, a) = 0]]$ is the occupancy measure of safe states visited by $\pi$.*

Proposition 3 illustrates that $c^*(s, a)$ misclassifies certain *safe* states as *unsafe* when their frequency in segments labeled *unsafe* exceeds that in segments labeled *safe* by the evaluator. We contend that this misclassification is likely to diminish with increased data collection or shorter segment lengths. However, it is important to note that this misclassification is guaranteed to be zero only when the segment length is reduced to one, meaning feedback is provided at the state level.

Additionally, note that the bias is non-negative, meaning that the expected cost $E_\pi[c_*(s, a)]$ acts as an upper bound on the true cost incurred by $\pi$. Therefore, ensuring that the policy does not exceed the threshold $c_{max}$ on $c_*$ guarantees that it adheres to the threshold on $c_{gt}$.

**Corollary 1.** *Any policy $\pi$ that is safe w.r.t $c_*$ is guaranteed to be safe w.r.t $c_{gt}$.*

In practice, we represent $p_\theta^{safe}(s, a)$ using a neural network with parameters $\theta$. The resulting cost function is defined as $c_\theta(s, a) \triangleq \mathbb{I}[p^{safe}(s, a)] < \frac{1}{2}$.

### 4.3 Efficient Subsampling of Trajectories

To reduce the burden on the evaluator and minimize the cost of feedback, we present a subset of the trajectories collected by the policy for feedback. The common approach is to break the problem into two parts: (1) define a schedule $N_{queries}(i)$ that determines the number of trajectories to be shown to the user at the end of each data collection round $i$. Subsequently, $N_{queries}(i)$ trajectories are sampled from those collected by the policy at data collection round $i$. While the ideal goal is to sample a subset of trajectories that maximizes the *expected value of information* [19], achieving this is computationally intractable [2]. To address this challenge, various sampling methods have been employed, seeking to maximize a surrogate measure of this value. Among these, *uncertainty sampling* stands out as the most prominent approach, wherein trajectories are sampled based on the estimator's uncertainty about their predictions [12, 18, 20]. However, quantifying the uncertainty is challenging given the lack of calibration in neural network predictions. To address this challenge, ensemble methods are frequently employed where the disagreement among the models is used as an uncertainty measure. However, the training of $n$ distinct neural networks can exact substantial resource costs, prompting consideration for alternative approaches.

In light of this, we introduce a new form of *uncertainty sampling* called *novelty sampling*. With *novelty sampling*, we gather all the *novel* trajectories after each round and present them to the evaluator for feedback. Formally, we define a state as novel if its density in the feedback data collected so far $d(s) = \sum_a d(s, a)$ is 0. A trajectory is deemed *novel* if it comprises of at least $e$ novel states. This can be interpreted as ensuring that the *edit distance*—a well-known measure of trajectory distance [27]—between the current trajectory and previously seen trajectories exceeds a threshold $e$. We do not consider novelty for state-action pairs as we found that extending to this case adversely impacted the performance.

The central notion is that the model is prone to errors on *novel* states- those it has not encountered during training. This arises because the policy evolves over time, venturing into states that were not previously encountered during data collection rounds. Therefore, this sampling strategy effectively reduces the *epistemic uncertainty* of the model—error arising from insufficient training data—thereby making it a form of *uncertainty sampling*. Furthermore, this sampling method offers the advantage of implicitly establishing a decreasing querying schedule as novelty of trajectories reduces over time as shown in Figure 6 in the Appendix.

We compute the density $d(s)$ through a count-based method, utilizing a hashmap to track the frequency of state occurrences in trajectories presented to the evaluator. Employing SimHash [10], we discretize the state space using a hash function $\phi : S \to \{-1, 1\}^n$, which maps *locally* similar states (measured by angular distance) to a binary code as:

$$\phi(s) = sgn(Ag(s)) \tag{7}$$

where $g : S \rightarrow \mathbb{R}^d$ is an optional prepossessing function and $A$ is an $n \times d$ matrix with i.i.d entries drawn from a standard normal distribution. Also, note that $n$ controls the granularity of the hash function, i.e, the number of states mapped to the same value. In our setting, $g(s) = s$ as we observed no improvement when employing functions like autoencoders for feature extraction.

## 4.4 Policy Optimization

After the data collection round, where we sample trajectories $\{\tau\}$ and their corresponding rewards $\{r(\tau)\}$ using $\pi$, we estimate their costs $c(\tau)$ using the inferred cost function. Our proposed method allows for the policy to be updated utilizing any on-policy constrained RL algorithm. In this study, we employ the PPO-Lagrangian algorithm[3] that combines the PPO algorithm [26] with a lagrangian multiplier to ensure safety.

A detailed description of the proposed method can be found in Algorithm 1. Lines [6-17] describe the data and feedback collection stage, and Lines [18-23] describe the cost inference and policy improvement stage.

---

**Algorithm 1** Reinforcement Learning from Safety Feedback (RLSF)

---

 1: **Input:** cost threshold $c_{max}$, segment length $k$, novelty criterion $e$
 2: **Initialize:** policy $\pi_0$
 3: **Initialize:** classifier $c_\theta$, learning rate $lr_\theta$ and feedback buffer $D$
 4: **Initialize:** $A \in \mathbb{R}^{n \times d}$ with entries sampled i.i.d from $\mathcal{N}(0,1)$, $\phi(.) = sgn(A^T(.))$, density map $d(.) \equiv 0$.
 5: **while** not converged **do**
 6:     Collect trajectories $\{\tau\}, \{r(\tau)\} \sim \pi$                 ▷ Data Collection
 7:     **for** each trajectory $\tau^i \in \{\tau\}$ **do**              ▷ Feedback Collection
 8:         novel $\leftarrow$ True if $\exists$ e states $\{s_e\} \in \tau^i$ such that $d(\phi(s_e)) > 0$
 9:         **if** novel **then**
10:             Show $\tau^i$ to the evaluator
11:             **for** each segment $\tau_{j:j+k-1} \in \tau^i$ **do**
12:                 Obtain feedback $y^{safe}$ for the segment $\tau^i_{j:j+k-1}$.
13:                 $D \leftarrow D \cup \{((s,a), y^{safe}) \;\; \forall (s,a) \in \tau^i_{j:j+k-1}\}$
14:                 $d(\phi(s)) \leftarrow d(\phi(s)) + 1 \;\; \forall s \in \tau^i_{j:j+k-1}$     ▷ Update the densities
15:             **end for**
16:         **end if**
17:     **end for**
18:     **for** each gradient step **do**               ▷ Update cost estimates
19:         Sample random minibatch $b \leftarrow \{(s,a), y_{safe}\} \sim D$
20:         $\theta \leftarrow \theta - lr_\theta \nabla L^{sur}(b)$
21:     **end for**
22:     Infer costs $\{c_\theta(\tau_i)\}$ for all $\tau^i \sim \{\tau\}$.
23:     Update $\pi$ using $\{r(\tau_i)\}$ and $\{c_\theta(\tau_i)\}$.        ▷ Policy Improvement
24: **end while**

---

## 5 Experiments

We investigate the following questions in our experiments:

1. Does RLSF succeed in effectively learning safe behaviours?

2. Can the inferred cost function be transferred across agents in the same task?

3. How does the proposed novelty based sampling scheme compare with other methods used in the literature?

4. How accurate is inferred cost function compared to the true cost function?

5. How can we address the overestimation bias of the inferred cost function as described in Section 4.2?

## 5.1 Experiment Setup

Table 1: Performance of different algorithms on the Safety Benchmarks. The first 7 environments represent the *hard* constraint case. The remaining environments illustrate the *soft* constraint case, with values in brackets indicating the cost threshold. Each algorithm is run for 6 independent seeds. (orange) and (blue) indicate the best performance in the known costs and inferred costs settings, respectively. Algorithms with a cost violation (C.V) rate below $1\%$ are deemed to have equal performance in terms of safety.

| Environment | | Cost Known (Best Run) | | Cost Inferred (Mean $\pm$ Standard error) | | |
|---|---|---|---|---|---|---|
| | | PPOLag | SIMKC | SDM | SIM | RLSF (Ours) |
| Point Circle | Return | 45.26 | **46.09** | $36.20 \pm 3.95$ | $22.26 \pm 9.59$ | $\mathbf{36.42 \pm 1.78}$ |
| | C.V Rate (%) | 0.4 | **0.43** | $11.43 \pm 0.69$ | $35.21 \pm 10.09$ | $\mathbf{1.9 \pm 0.09}$ |
| Car Circle | Return | **14.34** | 15.21 | $5.18 \pm 2.48$ | $6.34 \pm 2.87$ | $\mathbf{9.37 \pm 0.97}$ |
| | C.V Rate (%) | **0.84** | 5.4 | $6.2 \pm 6.18$ | $4.53 \pm 4.00$ | $\mathbf{0.54 \pm 0.30}$ |
| Biased Pendulum | Return | 717.43 | **983.27** | $495.58 \pm 160.84$ | $577.15 \pm 184.31$ | $\mathbf{721.48 \pm 111.49}$ |
| | C.V Rate (%) | 0.0 | **0.1** | $39.91 \pm 17.05$ | $48.58 \pm 21.67$ | $\mathbf{0 \pm 0}$ |
| Blocked Swimmer | Return | 22.62 | **21.05** | $86.96 \pm 10.69$ | $2.15 \pm 8.58$ | $\mathbf{16.09 \pm 1.44}$ |
| | C.V Rate (%) | 3.91 | **0.01** | $92.8 \pm 1.65$ | $13.33 \pm 12.11$ | $\mathbf{0.01 \pm 0.01}$ |
| HalfCheetah | Return | **2786.71** | 2497.82 | $3031.7 \pm 336.48$ | $257.34 \pm 147.35$ | $\mathbf{2112.63 \pm 161.26}$ |
| | C.V Rate (%) | **0.42** | 0.06 | $59.4 \pm 8.28$ | $0.0 \pm 0.0$ | $\mathbf{0.06 \pm 0.01}$ |
| Hopper | Return | **1705.00** | 1555.25 | $1097.57 \pm 56.35$ | $990.08 \pm 8.66$ | $\mathbf{1408.71 \pm 27.3}$ |
| | C.V Rate (%) | **0.19** | 0.02 | $0.0 \pm 0.0$ | $0.0 \pm 0.0$ | $\mathbf{0.29 \pm 0.02}$ |
| Walker2d | Return | **2947.25** | 2925.23 | $2195.94 \pm 134.21$ | $993.38 \pm 17.69$ | $\mathbf{2783.29 \pm 57.51}$ |
| | C.V Rate (%) | **0.16** | 0.0 | $1.58 \pm 1.53$ | $0.0 \pm 0.0$ | $\mathbf{0.05 \pm 0.01}$ |
| Point Goal | Return | **26.16** | 26.10 | $1.61 \pm 1.8149$ | $10.86 \pm 4.1$ | $\mathbf{24.65 \pm 0.59}$ |
| | Cost (40.0) | **34.19** | 31.83 | $30.57 \pm 13.29$ | $52.76 \pm 12.85$ | $\mathbf{35.08 \pm 1.08}$ |
| Car Goal | Return | 27.37 | **26.44** | $1.05 \pm 2.83$ | $\mathbf{10.88 \pm 7.1}$ | $24.28 \pm 2.1$ |
| | Cost (40.0) | 41.67 | **35.41** | $34.71 \pm 9.87$ | $\mathbf{33.33 \pm 11.26}$ | $41.25 \pm 2.27$ |
| Point Push | Return | 6.00 | **10.84** | $0.16 \pm 0.14$ | $3.63 \pm 1.77$ | $\mathbf{2.68 \pm 1.03}$ |
| | Cost (35.0) | 26.08 | **26.96** | $22.89 \pm 5.95$ | $45.43 \pm 3.86$ | $\mathbf{30.51 \pm 3.4}$ |
| Car Push | Return | **3.07** | 2.68 | $-3.04 \pm 3.3$ | $1.56 \pm 0.46$ | $\mathbf{1.54 \pm 0.51}$ |
| | Cost (35.0) | **20.53** | 20.95 | $23.25 \pm 7.78$ | $36.55 \pm 1.48$ | $\mathbf{27.69 \pm 1.19}$ |

We evaluate RLSF on multiple continuous control benchmarks in the Safety Gymnasium environment [17] and Mujoco [30] based environments introduced in [22]. The *Circle*, *Blocked Swimmer* and *Biased Pendulum* environments constrain the position of the agent whereas the *Half Cheetah*, *Hopper* and *Walker-2d* environments constrain the velocity of the agent. The *Goal* and *Push* tasks are the most challenging as they contain static and dynamic obstacles that the agent must avoid while completing the task. All of the above environments reflect safety challenges that an agent could potentially face in real-world scenarios. Additionally, we conduct experiments using the Driver simulator introduced in [21], which presents two scenarios that an autonomous driving agent is likely to encounter on the highway: lane changes and blocked paths. In this setup, the cost function is based on multiple variables, including speed, position, and distance to other vehicles. Additionally, we introduce a third scenario: overtaking on a two-lane highway. A detailed description of the tasks can be found in the Appendix B.

We split the environments into two settings: (1) *hard constraint* setting ($c_{max} = 0$) where the safety of the policy is measured in terms of the cost violation (CV) rate defined as the number of cost violations divided by the length of the episode and (2) *soft* constraint setting where $c_{max} > 0$. We utilize an automated script that leverages the underlying cost function to simulate the feedback provided by the evaluator.

We compare the performance of our algorithm against the following baselines: **Self Imitation Safe Reinforcement Learning (SIM)** [16]: SIM is a state-of-the-art method in constrained RL that also supports the case where feedback is elicited from an external evaluator. Similar to RLSF, the method consists of two stages, a data collection/feedback stage and a policy optimization stage. In the first stage, a trajectory $\tau$ is labelled as *good* if $[r(\tau) \geq r_{good} \ \cap \ \mathbb{I}[c(\tau) \leq c_{max}]$, and labelled as *bad* if $[r(\tau) \geq r_{bad} \ \cup \ \mathbb{I}[c(\tau) \geq c_{max}]$, where $r_{good}$ and $r_{bad}$ are predefined thresholds on the reward. The information $\mathbb{I}[c(\tau) \leq c_{max}]$ is received from feedback. The idea is then to imitate the *good*

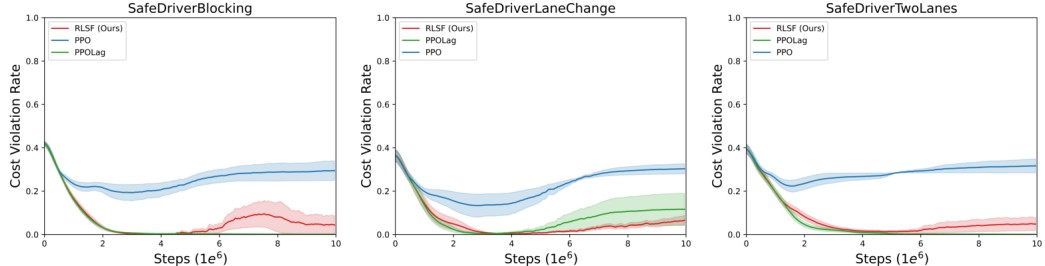

Figure 1: Cost Violation rate of different algorithms in the Driver environment. Each algorithm is run for 6 independent seeds, with the curves representing the mean and the shaded regions indicating the standard error.

trajectories and stay away from the *bad* trajectories. If $\rho^\pi$ is the occupancy measure of the current policy $\pi$ and $\rho^G, \rho^B$ are the occupancy measures of the *good* and *bad* trajectories respectively, then, $\pi$ is optimized as,

$$\pi^* = \underset{\pi}{\operatorname{argmax}} \ \text{KL}(\rho^{\pi,G}||\rho^B) \tag{8}$$

where $\rho^{\pi,G} = (\rho^\pi + \rho^G)/2$ and KL denotes the Kullback–Leibler divergence.

**Safe Distribution Matching (SDM)**: SIM combines rewards with safety feedback into a joint notion of *good* and *bad* which may not be desirable when the cost function is unknown. Thus, we introduce an additional baseline that keeps these two signals separate by labelling the trajectory as *good* if $c(\tau) \leq c_{max}$ and *bad* otherwise. The policy is then updated as,

$$\pi^* = \underset{\pi}{\operatorname{argmax}} \ r + \lambda \text{KL}(\rho^{\pi,G}||\rho^B) \tag{9}$$

where $\lambda \in [0,1]$ controls the tradeoff between the two objectives.

As an upper bound, we compare the performance of our algorithm to scenarios where the cost function is known: PPO-Lagrangian (PPOLag) [3] and SIM with known costs (SIMKC) [16]. The objective is not to surpass their performance, but to match it. Therefore, when reporting the results of these algorithms, we present the best-performing seed across multiple runs.

All results presented use novelty-based sampling unless stated otherwise. We use a segment length of 1 in the *Driver*, *Goal* and *Push* environments. This is because the Safety Goal and Push environments contain small obstacles that the agent interacts with for very brief periods of time, hence requiring more fine-grained feedback. An example of this is present in Figure 11 in the Appendix. In the *Driver* environments, a randomly initialized policy was *highly unsafe*. Thus a long segment length would force the evaluator to label every segment *unsafe*, making cost inference infeasible. **In the remaining environments, the segment length corresponds to the length of the episode.** Details on the number of queries generated for feedback can be found in Table 4 in the Appendix. We grant the two baseline methods (with unknown costs) an advantage by providing feedback for every trajectory generated.

## 5.2 Cost Inference across various tasks

**Benchmark Environments** Table 1 presents the performance of the various algorithms on the benchmark environments. RLSF significantly outperforms the two baselines in terms of reward and safety in *all* of the environments [2]. RLSF comes to within $\approx 80\%$ of the performance of the best run of PPOLag in 7/11 environments thereby underscoring its effectiveness in learning safe policies.

**Driving Scenarios** In the driving environments, we evaluate the performance of our algorithm against two baseline methods: a naive PPO policy that solely maximizes reward and the PPO-Lag algorithm. Figure 1 presents these results. Notably, our method demonstrates comparable safety

---
[2]except the *Car Goal* environment where RLSF and PPOLag marginally exceed the threshold

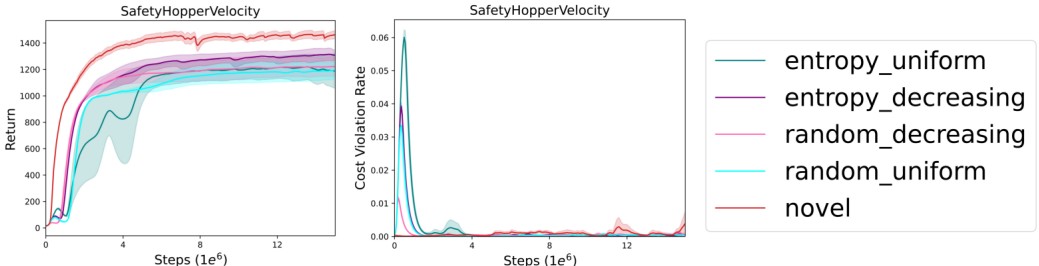

Figure 2: Comparison of different sampling and scheduling schemes. Results are averaged over 3 independent seeds. The proposed sampling method generates on average $\approx 1950$ queries, hence for fair comparison the other methods were given a budget of 2000 queries.

performance to PPO-Lag while significantly outperforming PPO in terms of cost violations. This highlights the effectiveness of our approach in learning safe autonomous driving policies. Video clips showcasing the learned policies in the 3 scenarios are included in the supplementary material.

### 5.3 Cost Transfer

We also showcase the potential of using the inferred cost function to train an agent with a different embodiment or morphology to solve the same task from scratch, without requiring any additional feedback. We utilize the inferred cost function obtained during the training of a *Point* agent to train the *Doggo* agent for the *Circle* and *Goal* tasks. The observation space in the environments consists of agent-related telemetry (acceleration, velocity) and task-related information (lidar sensors for goal/boundary/obstacle detection). In these experiments, we do not incorporate agent-related information when learning the cost function, ensuring that it can be transferred to a new agent. Next, the inferred cost function remains fixed during the training of the second agent, and the task-related features are utilized for cost inference. We compare the performance of this approach with that of the PPO-Lagrangian algorithm that utilizes the underlying cost function of the environment. From Table 2 we can infer that agents trained using transferred cost function are comparable in performance to agents trained using the *true* underlying cost function.

Table 2: Comparison of PPOLag performance when trained with the underlying task cost function versus the transferred cost function. Results are averaged over three independent seeds.

| Source Env | Target Env | | PPOLag with true cost | PPOLag with transferred cost |
|---|---|---|---|---|
| Point Circle | Doggo Circle | Return | $2.69 \pm 0.24$ | $2.00 \pm 0.27$ |
| | | C.V Rate (%) | $0.63 \pm 0.09$ | $0.18 \pm 0.05$ |
| Point Goal | Doggo Goal | Return | $1.45 \pm 0.06$ | $1.20 \pm 0.26$ |
| | | Cost (40.0) | $40.86 \pm 4.80$ | $37.31 \pm 6.77$ |

### 5.4 Effect of Novelty Sampling

Next, we demonstrate the effectiveness of the proposed *novelty-based sampling* mechanism by comparing it with other popular querying methods from the literature [18, 20]. These methods require a predefined budget for the number of trajectories presented to the evaluator. We evaluate two schedules: (1) *Uniform schedule*, where a fixed number of trajectories is shown for feedback after each round, and (2) *Decreasing schedule*, where the number of trajectories decreases in proportion to $\frac{1}{t}$ each round. For each schedule, we test the following strategies for sampling the subset of trajectories to be presented to the evaluator: (a) *Random sampling*, which selects a random subset of trajectories, and (b) *Entropy sampling*, where trajectories are sampled in descending order of the average entropy $\mathcal{H}(p_\theta^{safe}(s_t, a_t))$ of the estimator. As shown in Figure 2, entropy based sampling outperforms random sampling, and a decreasing schedule outperforms a uniform schedule. However, all the methods fall short compared the proposed *novelty based sampling* mechanism.

## 5.5 Analyzing the Inferred Cost function

To further evaluate the quality of the inferred reward function, we plot it alongside the true underlying cost function of the environment as presented in Figure 3. In the Point Goal environment, where feedback is collected for each state, the inferred cost function closely aligns with the true cost given sufficient data. However, in the Point Circle environment, where feedback is collected at the trajectory level, the inferred cost function exhibits a slight overestimation bias, as discussed in Section 4.2. Nevertheless, from Figure 3, we can conclude that our method learns cost functions that strongly correlate with the true costs.

## 5.6 Bias Correction

In Section 4.2, we established that when feedback is collected over segments, the inferred cost function tends to exhibit an overestimation bias. As a result, optimizing for a policy that is deemed *safe* with respect to $c_{max}$ using the inferred cost function can lead to overly conservative policies. If the bias $b$, could be calculated apriori, then adjusting the safety threshold to $c_{max} + b$ would still satisfy the safety guarantees outlined in Corollary 1. However, since estimating $b$ in advance is infeasible, we propose a heuristic approach where a constant $\delta \in \mathbb{R}^+$ is added to $c_{max}$ to account for the bias. It's important

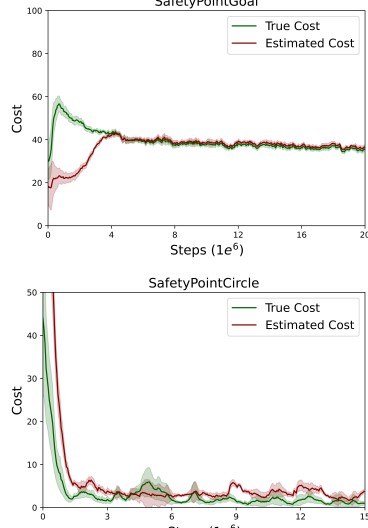

Figure 3: Comparing the inferred cost to the true cost.

to note that Corollary 1 only holds if $\delta \leq b$, meaning this heuristic doesn't guarantee that the resulting policy will always be safe. Nevertheless, we tested this approach with different $\delta$ values in the *Car Circle* environment, as this setting exhibited a higher overestimation bias (as reflected in Table 1). The results, shown in Figure 4, demonstrate that adding this bonus does improve performance (with higher $\delta$ values yielding better results), though it introduces the additional challenge of tuning $\delta$.

# 6 Limitations, Future Work and Broader Impact

In this work, we consider the problem of learning cost functions from feedback in scenarios where evaluating or defining such functions is either costly or infeasible. We present a novel framework which significantly reduces the burden on the evaluator by eliciting feedback over extended horizons, a challenge that has not been addressed in previous research. To further reduce their load, we propose a *novelty-based sampling* method that only presents previously unseen trajectories to the evaluator. Through experiments on multiple benchmark environments, we demonstrate the effectiveness of our framework and the proposed sampling algorithm in learning safe policies. However, there are a few limitations we would like to acknowledge. First, our method relies on state-level feedback in some environments, which can be expensive to obtain. Second, while we simulate feedback using the true cost function, real-world feedback is often noisy, especially when provided by human evaluators. It would be valuable to conduct experiments involving real human subjects to validate the approach.

By enabling safer autonomous systems, our approach has the potential to enhance safety across a range of applications, including autonomous vehicles, drones used for delivery, and industrial robots operating in environments like warehouses or manufacturing plants. These improvements could significantly reduce accidents, protect human operators, and increase the overall reliability of automated systems.

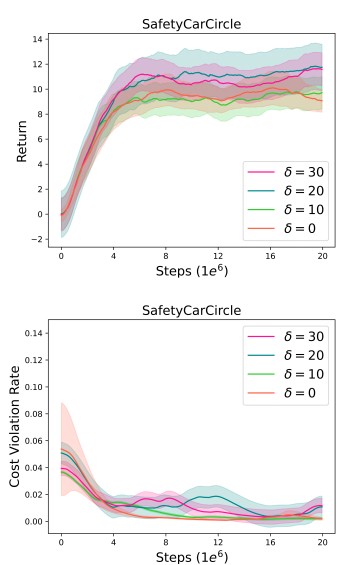

Figure 4: Increasing $c_{max}$ by $\delta$ to correct for the overestimation bias.

## 7    Acknowledgments

This research/project is supported by the National Research Foundation Singapore and DSO National Laboratories under the AI Singapore Programme (AISG Award No: AISG2-RP-2020-016) and Lee Kuan Yew Fellowship awarded to Pradeep Varakantham.

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

# A Proofs

**Proposition 1.** *The surrogate loss $L^{sur}$ is an upper bound on the likelihood loss $L^{mle}$.*

*Proof.* We have,

$$L^{sur} - L^{mle} = -E_{(\tau_{i:j}, y^{safe}) \sim P}(1 - y^{safe}) \left[ \sum_{t=i}^{j} \log\left(1 - p^{safe}(s_t, a_t)\right) \right.$$

$$\left. - \log\left(1 - \prod_{t=i}^{j} p^{safe}(s_t, a_t)\right) \right]$$

$$= E_{(\tau_{i:j}, y^{safe}) \sim P}(1 - y^{safe}) \left[ \log\left( \frac{1 - \prod_{t=i}^{j} p^{safe}(s_t, a_t)}{\prod_{t=i}^{j} 1 - p^{safe}(s_t, a_t)} \right) \right]$$

Notice that the term,

$$\log\left( \frac{1 - \prod_{t=i}^{j} p^{safe}(s_t, a_t)}{\prod_{t=i}^{j} 1 - p^{safe}(s_t, a_t)} \right) \geq 0$$

This is because

$$1 - \prod_{t=i}^{j} p^{safe}(s_t, a_t) \geq \prod_{t=i}^{j} 1 - p^{safe}(s_t, a_t)$$

which we prove subsequently.

**Lemma 1.** *For any sequence of numbers $\{x_1, \ldots, x_n\}$, where $x_i \in [0, 1]$ for all $i \in 1, \ldots, n$ we have $1 - \prod_{i=1}^{n} x_i \geq \prod_{i=1}^{n}(1 - x_i)$.*

*Proof.* Using the AM-GM inequality we have,

$$\sqrt[n]{\prod_{i=1}^{n} x_i} \leq \frac{\sum_{i=1}^{n} x_i}{n}$$

$$1 - \prod_{i=1}^{n} x_i \geq 1 - \left( \frac{\sum_{i=1}^{n} x_i}{n} \right)^n \geq 1 - \left( \frac{\sum_{i=1}^{n} x_i}{n} \right) \tag{10}$$

Similarly, we have,

$$\sqrt[n]{\prod_{i=1}^{n}(1 - x_i)} \leq \frac{\sum_{i=1}^{n}(1 - x_i)}{n}$$

$$-\prod_{i=1}^{n}(1 - x_i) \geq -\left( \frac{\sum_{i=1}^{n}(1 - x_i)}{n} \right)^n \geq -\left( \frac{\sum_{i=1}^{n}(1 - x_i)}{n} \right) \tag{11}$$

Adding Eq 10 and Eq 11, we get,

$$1 - \prod_{i=1}^{n} x_i - \prod_{i=1}^{n}(1 - x_i) \geq 1 - \left( \frac{\sum_{i=1}^{n} x_i}{n} \right) - \left( \frac{\sum_{i=1}^{n}(1 - x_i)}{n} \right)$$

$$1 - \prod_{i=1}^{n} x_i - \prod_{i=1}^{n}(1 - x_i) \geq 0$$

$$1 - \prod_{i=1}^{n} x_i \geq \prod_{i=1}^{n}(1 - x_i) \tag{12}$$

$\square$

$\square$

**Proposition 2.** *The optimal solution to Eq 4 yields the estimate,*

$$p_*^{safe}(s,a) = \frac{d_g(s,a)}{d_g(s,a) + d_b(s,a)}$$

*Proof.* The proof is obtained by differentiating Eq 4 w.r.t $p^{safe}(s,a)$ and setting it to 0. $\square$

**Proposition 3.** *For a fixed policy $\pi$, the bias in the estimation of the incurred costs is given by,*

$$E_\pi[c_*(s,a)] - E_\pi[c_{gt}(s,a)] = E_{(s,a)\sim d_g^\pi}[\mathbb{I}[d_b > d_g]]$$

*where $\rho_g^\pi(s,a) = E_\pi[\sum_{t=0}^{T}[\mathbb{I}[(s_t,a_t) = (s,a) \cap c_{gt}(s,a) = 0]]$ is the occupancy measure of true safe states visited by $\pi$.*

*Proof.* The estimated cost function $c_*(s,a) = \mathbb{I}[p_*^{safe}(s,a) < \frac{1}{2}]$, from Eq 5 we get $p_*^{safe}(s,a) < \frac{1}{2}$ when $d_b(s,a) > d_g(s,a)$. Thus, $c_*(s,a) = \mathbb{I}[d_b(s,a) > d_g(s,a)]$.

$$E_\pi[c_*(s,a) - c_{gt}(s,a)] = E_{\tau\sim\pi} \sum_{t=0}^{T} [c_*(s_t,a_t) - c_{gt}(s_t,a_t)]$$

$$= E_{\tau\sim\pi} \sum_{t=0}^{T} \sum_{s,a} \mathbb{I}[(s_t,a_t) = (s,a)][c_*(s_t,a_t) - c_{gt}(s_t,a_t)]$$

$$= E_{\tau\sim\pi} \sum_{t=0}^{T} \sum_{s,a} \Big[ \mathbb{I}[(s_t,a_t) = (s,a) \cap c_{gt}(s,a) = 0] \big[c_*(s_t,a_t) - c_{gt}(s_t,a_t)\big]$$

$$+ \mathbb{I}[(s_t,a_t) = (s,a) \cap c_{gt}(s,a) = 1] \big[c_*(s_t,a_t) - c_{gt}(s_t,a_t)\big] \Big]$$

$$= E_{\tau\sim\pi} E_{(s,a)\sim\rho_g^\pi}[c_*(s,a) - 0] + E_{(s,a)\sim\rho_b^\pi}[c_*(s,a) - 1]$$

where $\rho_g^\pi(s,a) = E_{\tau\sim\pi}[\sum_{t=0}^{T}[\mathbb{I}[(s_t,a_t) = (s,a) \cap c_{gt}(s,a) = 0]]$ and $\rho_b^\pi(s,a) = E_{\tau\sim\pi}[\sum_{t=0}^{T}[\mathbb{I}[(s_t,a_t) = (s,a) \cap c_{gt}(s,a) = 1]]$.

Note that when a state is *unsafe*, i.e, $c_{gt}(s,a) = 1$, then, $p_*^{safe}(s,a) = 0$ as $n_g(s,a) = 0$. Thus for all of these states, $c_*(s,a) = 1$. Hence, we are left with,

$$E_\pi[c_*(s,a) - c_{gt}(s,a)] = E_{(s,a)\sim\rho_g^\pi}[c_*(s,a)]$$

$$= E_{(s,a)\sim\rho_g^\pi}[\mathbb{I}[d_b > d_g]]$$

$\square$

**Corollary 2.** *Any policy $\pi$ that is safe w.r.t $c_*$ is guaranteed to be safe w.r.t $c_{gt}$.*

*Proof.* From Proposition 3, we know that $E_\pi[c_*(s,a)] - E_\pi[c_{gt}(s,a)] = E_{(s,a)\sim\rho_g^\pi}[\mathbb{I}[d_b > d_g]]$. Since $E_{(s,a)\sim\rho_g^\pi}[\mathbb{I}[d_b > d_g]] \geq 0$, we have $E_\pi[c_*(s,a)] \geq E_\pi[c_{gt}(s,a)]$.

Thus, $E_\pi[c_*(s,a)] \leq c_{max} \implies E_\pi[c_{gt}(s,a)] \leq c_{max}$. $\square$

## B  Environments

### B.1  Benchmark environments:

**Mujoco** [30]-based environments extend traditional locomotion tasks by incorporating safety constraints. These environments are categorized into two types: position-based and velocity-based constraints. A detailed description of each is provided below:

1. **Position Based:** These environments, recently introduced as benchmarks in [22], are explained in more detail below.

(a) *Blocked Swimmer*: The agent controls a robot composed of three segments joined at two points. The objective is to move the robot to the right as quickly as possible by applying torque through rotors at the joints. The agent receives a reward at each time step, proportional to its displacement in the $X$-direction. The episode concludes after 1000 time steps. To increase the challenge, a cost is imposed when the agent's $X$-coordinate exceeds 0.5, constraining it to the region $-\infty \leq X < 0.5$.

(b) *Biased Pendulum*: The agent's goal is to balance a pole on a cart. Each episode ends if the pole falls or after a maximum of 1000 time steps. The agent receives a reward of 1 when its $X$-coordinate is within the range $(-\infty, -0.01]$, which monotonically decreases to 0.1 as the $X$-coordinate approaches 0. Beyond this, the agent receives a constant reward of 0.1, encouraging movement in the leftward direction. To prevent excessive leftward movement, a cost of 1 is incurred when the $X$-coordinate is in the region $(-\infty, -0.015]$, constraining the agent from moving too far left.

2. **Velocity Based:** These environments are part of the Safety Gymnasium benchmarks, introducing a velocity constraint for the *Half-Cheetah*, *Hopper*, and *Walker-2d* agents. The agent incurs a cost of 1 whenever it exceeds a velocity threshold of $v_{max}/2$, where $v_{max}$ is the maximum velocity achieved by the agent when trained with PPO for $1 \times 10^7$ steps.

**Safety Gymnasium** [17] has emerged as a prominent benchmark for evaluating constrained re-inforcement learning algorithms, providing a challenging framework for cost inference. The environments feature multiple agents, including the *Point*, *Car*, and *Doggo*, listed in increasing order of difficulty in controlling the agent. The objective is to complete specific tasks using one of these agents.

In the *Circle* task, the agent must navigate around the boundary of a circle centered at the origin while remaining within a safe region defined by two vertical boundaries at $x = \pm 0.7$. A cost of 1 is incurred each time the agent exceeds this boundary ($|x| \geq 0.7$).

In the *Goal* task, the agent must reach a designated goal area as quickly as possible while avoiding both static and dynamic obstacles, incurring a cost of 1 for each collision.

The *Push* task builds upon the *Goal* task by adding the challenge of pushing a block to the goal area while still avoiding collisions with obstacles in the environment.

## B.2 Safe Driver

Driving-based environments are emerging as key benchmarks for addressing the problem of constraint inference [22], particularly due to the complex safety considerations involved in driving. We evaluate our algorithm using the *Driver* simulator introduced in [21, 24] to learn safe behaviors from feedback. This environment encompasses two scenarios that a self-driving agent is likely to encounter on the highway: a blocked road and a lane change. Additionally, we introduce a third scenario that involves overtaking on a two-lane highway, where the agent must safely pass a slower car in its lane by utilizing the second lane. This maneuver poses risks, as traffic in this lane is moving in the opposite direction, necessitating careful execution. Below, we provide a brief description of the environment.

The Driver environment utilizes point-mass dynamics and features continuous state and action spaces. The state of a vehicle in the simulation is represented by the tuple $(x, y, \phi, v)$, which includes the agent's position $(x, y)$, heading $\phi$ and velocity $v$. The observation of the ego vehicle is formed by concatenating the states of all vehicles in the scene. The environment allows for two actions $(a_1, a_2)$ where $a_1$ represents the steering input $a_2$ denotes the applied acceleration.

The state of the agents evolves as,

$$s_{t+1} = (x + \delta x, y + \delta y, \phi + \delta \phi, v + \delta v)$$
$$(\delta x, \delta y, \delta \phi, \delta v) = (v \cos \phi, v \sin \phi, a_2 - \alpha v)$$

where $\alpha$ is used to control the friction. Additionally, the velocity is clipped to the range $[-1, 1]$ at each timestep.

The task is to control the *ego* vehicle (white in Figure 5) to reach the top of the road as soon as possible while adhering to costs on the velocity and proximity to other vehicles. The trajectories of the other vehicles are fixed, with some random noise applied to their steering and acceleration inputs.

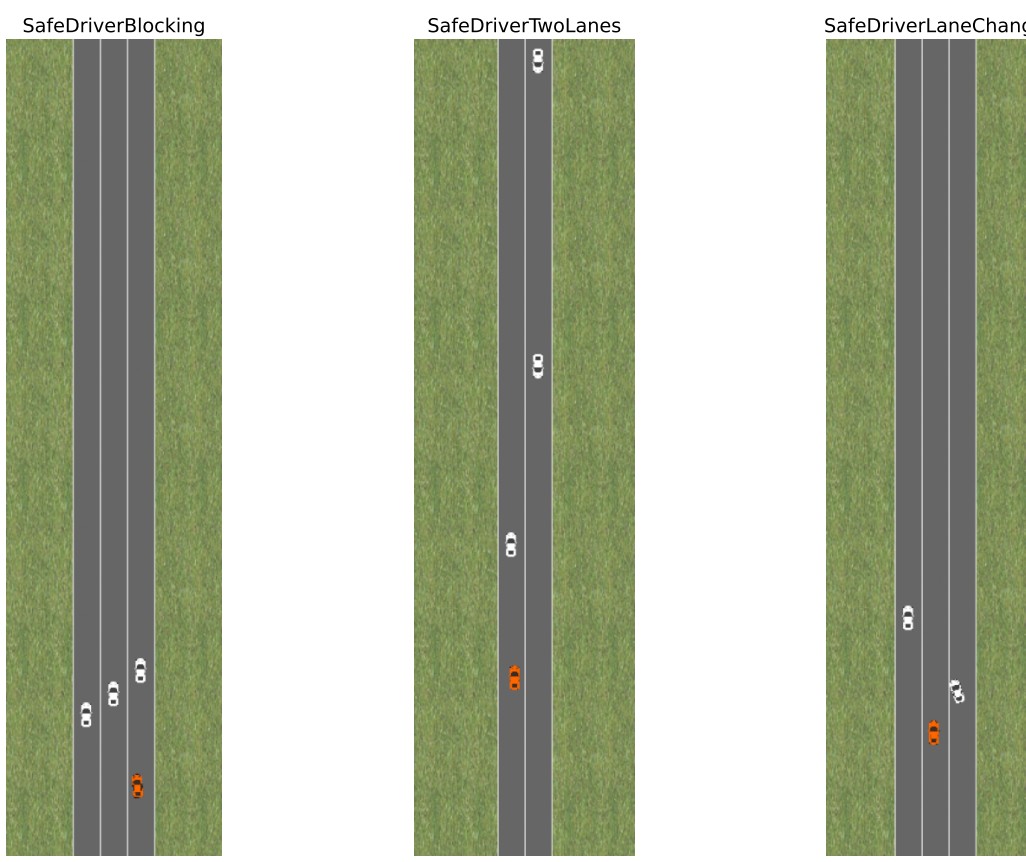

Figure 5: Driver Environments

The agent receives a reward $r_t = 10(y_t - y_{t-1})$ at each time step, incentivizing it to navigate through traffic rapidly. Additionally, it incurs a penalty of $-1$ for going off the road, driving backward, or failing to stay in the center of the lane. A collision results in a penalty of $-100$. The episode terminates when the agent reaches the top of the road or collides with another vehicle. Although these constraints can be integrated into the reward function, they can still lead to unsafe behaviors, highlighting the necessity for an additional cost function. The PPO agent is trained using this reward function.

In the CMDP settings, the cost function incorporates some of the constraints from the reward function. For instance, the agent incurs a cost of $1$ when it goes off the road or drives backward. Additional costs are incurred when the agent crosses the speed limit $v_{max}$ or gets too close to any nearby vehicles. The *ego* car is deemed to close to a neighbouring vehicle if $\exp -b(c_1 d_x^2 + c_2 d_y^2) + ba \geq 0.4$, where $a = 0.01$, $b = 30$, $c_1 = 10$, $c_2 = 2$ and $d_x$ and $d_y$ represent the distance of the nearby vehicle from the *ego* vehicle.

## C Related Work

**Constrained RL**  The Constrained Markov Decision Process (CMDP) framework [4] has emerged as a valuable tool in safety-critical settings, offering a clear separation between task-related information (rewards) and safety-related information (costs). This separation simplifies environment specification and allows for easier transfer of cost functions across similar environments. While previous research has extensively addressed this problem [1, 35, 34, 16], these studies typically assume that the cost function is known. However, this assumption can be limiting when designing the cost function is complex or costly to evaluate. Consequently, there is a need to infer the cost function from external data in such cases.

**Cost Inference**   Historically, cost inference has primarily depended on two types of data: (1) evaluator feedback and (2) constraint abiding demonstrations obtained from an expert.

Previous work on learning from feedback often relies on limiting assumptions such as deterministic transitions and smoothness [28, 32, 11] or linearity assumptions [5] on the cost function. Moreover, none of these approaches address the problem of actively querying humans to minimize the required feedback. Although this has been theoretically explored in [8], the proposed algorithm has not yet been adapted for complex environments. Additionally, these methods are typically limited to feedback on individual state-action pairs, which can be expensive when relying on human evaluators. Our RLSF approach addresses these limitations by reducing the amount of feedback required and avoiding assumptions about the cost function.

Inferring cost functions from constraint-satisfying expert demonstrations has gained significant attention recently [6, 33, 22]. However, acquiring such expert demonstrations can be challenging in many scenarios. An alternative form of feedback is human interventions. For instance, [23] rely on the human evaluator to terminate the episode when the agent is about to engage in unsafe behavior, while [25] depend on the human to take control of the agent in such situations and guide it to a safe state.

**Novelty-Based Sampling**   Novelty estimation has been widely explored in the context of active exploration in reinforcement learning. The connection between prediction accuracy and state novelty has been extensively studied, as highlighted by [9]. Additionally, state density estimation has been investigated in works such as [29, 7, 13], where infrequently visited states are assigned an exploration bonus. Notably, [29] also uses Simhash [10] to maintain state density estimates.

**Reward Inference**   This work also connects to the analogous problem of reward inference in standard MDPs, which similarly rely on human feedback [12, 18, 20] and expert demonstrations [15, 14].

# D   Experiments

## D.1   Hyperparameters

We conducted the experiments on a cluster quipped with 4 NVIDIA RTX A5000 GPUs and 96 core CPUs. The experiments took an approximate of two weeks to run. The detailed hyperparameters utilized in the experiments are presented in Table 3. All results presented in both the main paper and the appendix are based on three independent seeds, with the mean and standard error reported, unless explicitly stated otherwise.

Table 3: Hyper Parameters

| Hyper Parameter | Point Circle | Car Circle | Biased Pendulum | Blocked Swimmer | Half Cheetah | Hopper | Walker 2d | Point/Car Goal | Point/Car Push | Safe Driver |
|---|---|---|---|---|---|---|---|---|---|---|
| Actor hidden size | [256, 256, 256] | [256, 256, 256] | [256, 256, 256] | [256, 256, 256] | [256, 256, 256] | [256, 256, 256] | [256, 256, 256] | [256, 256, 256] | [256, 256, 256] | [64] |
| (Value/Cost) Critic hidden size | [256, 256, 256] | [256, 256, 256] | [256, 256, 256] | [256, 256, 256] | [256, 256, 256] | [256, 256, 256] | [256, 256, 256] | [256, 256, 256] | [256, 256, 256] | [64] |
| Classifier Network | [64,64] | [64,64] | [32] | [32] | [32] | [32] | [32] | [64, 64] | [64, 64] | [64] |
| Gamma | 0.99 | 0.99 | 0.99 | 0.99 | 0.99 | 0.99 | 0.99 | 0.99 | 0.99 | 0.99 |
| lr (Actor/Critics/Classifier) | 0.0001 | 0.0001 | 0.0001 | 0.0001 | 0.0001 | 0.0001 | 0.0001 | 0.0001 | 0.0001 | 0.0001 |
| lr Lagrangian | 0.01 | 0.01 | 0.01 | 0.01 | 0.01 | 0.01 | 0.01 | 0.01 | 0.01 | 0.01 |
| n_epochs (PPO/Critics) | 160 | 160 | 160 | 160 | 160 | 160 | 160 | 160 | 160 | 160 |
| n_epochs Classifier | 5000 | 5000 | 5000 | 5000 | 5000 | 5000 | 5000 | 5000 | 5000 | 5000 |
| Classifier batch size | 4096 | 4096 | 4096 | 4096 | 4096 | 4096 | 4096 | 4096 | 4096 | 4096 |
| SimHash Embedding size (k) | 11 | 13 | 64 | 16 | 15 | 15 | 15 | 16 | 16 | 24 |

## D.2   Main Experiments

The training curves for the benchmark experiments and the *Driver* environment are presented in Figure 7 and Figure 8. Additionally information on the number of queries presented for feedback is presented in Table 4.

## D.3   Novelty based Sampling

Figure 6 shows the implicit decreasing schedule encountered when using *novelty based sampling* in various environments. Figure 9 shows additional comparison with other querying mechanisms in the Biased Pendulum environment. To further assess the effectiveness of *novelty based sampling*, we categorized the trajectories collected during a data collection round based on novelty. We then

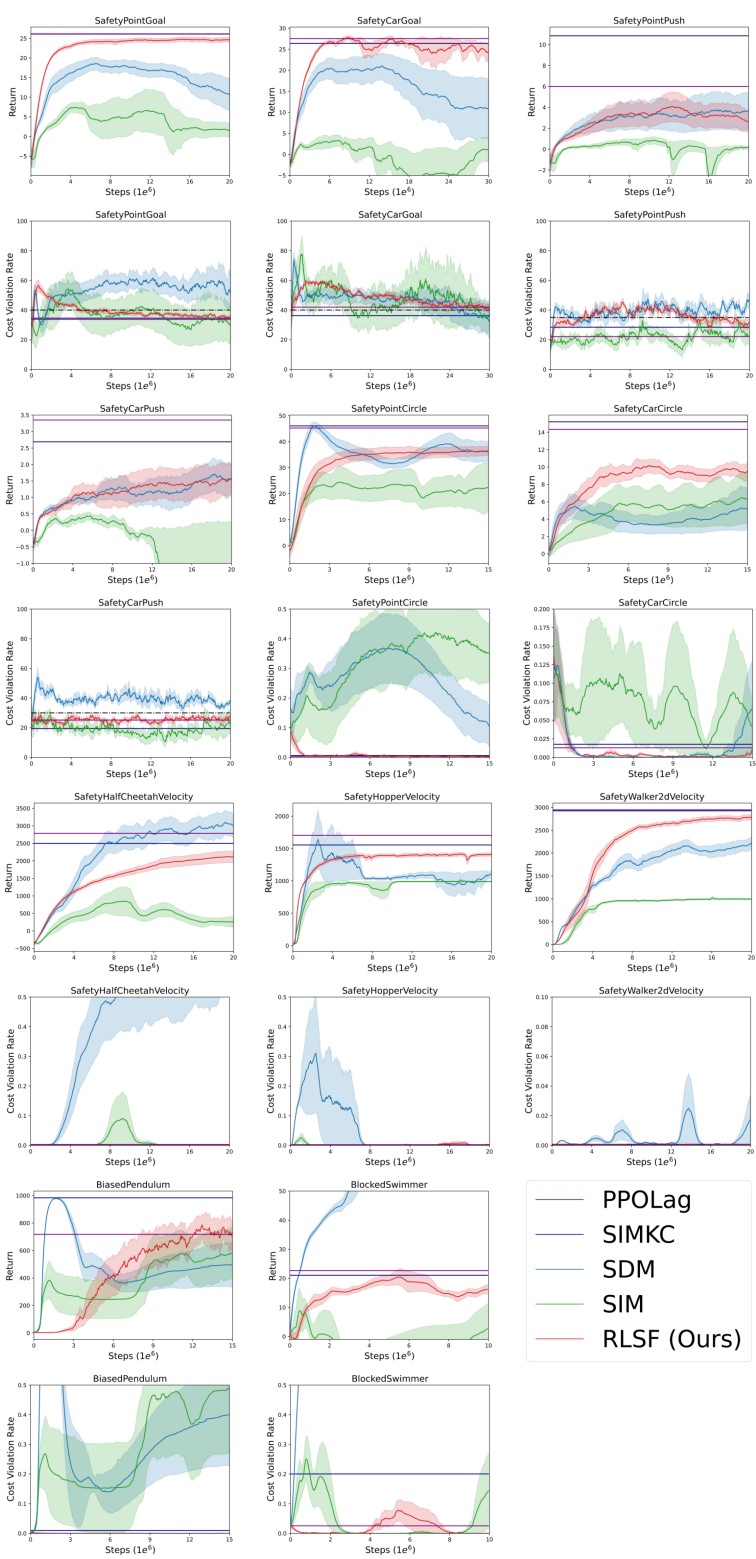

Figure 7: Training curves depicting the performance of different algorithms, based on six independent seeds. The curves show the mean returns, with shaded regions representing the standard error. Parallel lines indicate the best-performing run.

Table 4: Number of trajectories presented to the evaluator for feedback

| Nature of Costs | Environment | Number of Trajectories shown | Episode Length | Segment Length ($k$) |
|---|---|---|---|---|
| Position | Point Circle | $889.66 \pm 185.27$ | 500 | 500 |
| | Car Circle | $1063 \pm 73.48$ | 500 | 500 |
| | Biased Pendulum | $3939.33 \pm 231.89$ | 1000 | 1000 |
| | Blocked Swimmer | $646.83 \pm 127.56$ | 1000 | 1000 |
| Velocity | HalfCheetah | $4112.83 \pm 134.26$ | 1000 | 1000 |
| | Hopper | $1924.99 \pm 92.59$ | 1000 | 1000 |
| | Walker2d | $4170.33 \pm 174.83$ | 1000 | 1000 |
| Obstacles | Point Goal | $7892 \pm 178.15$ | 1000 | 1 |
| | Car Goal | $3866.5 \pm 80.83$ | 1000 | 1 |
| | Point Push | $6409.33 \pm 77.46$ | 1000 | 1 |
| | Car Push | $3368.33 \pm 176.36$ | 1000 | 1 |
| Driving | Blocking | $6242.3 \pm 778.13$ | 100 | 1 |
| | Two Lanes | $2912.37 \pm 443.31$ | 100 | 1 |
| | Lane Change | $3510.82 \pm 644.28$ | 100 | 1 |

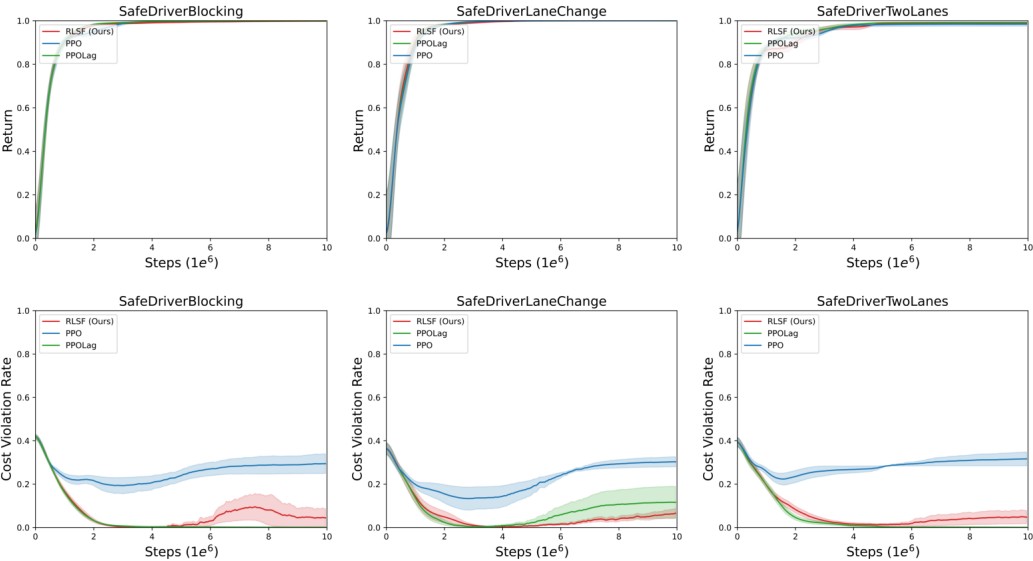

Figure 8: Performance of algorithms in different driving scenarios. Each algorithm is run for 6 independent seeds, curves represent the mean and shaded regions represent the standard error. Normalized return is given by the formula $r = \frac{r - r_{\text{PPO}}}{r - r_{\text{random}}}$ where $r_{\text{PPO}}$ is the return of PPO and $r_{\text{random}}$ is the return of a random policy.

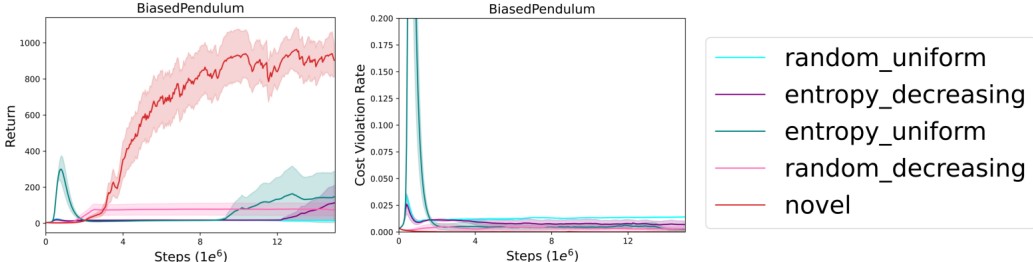

Figure 9: Comparison of different sampling and scheduling schemes. The results are averaged over 3 independent seeds. The proposed sampling method generates on average 3600 queries, hence for fair comparison the other methods were given a budget of 4000 queries.

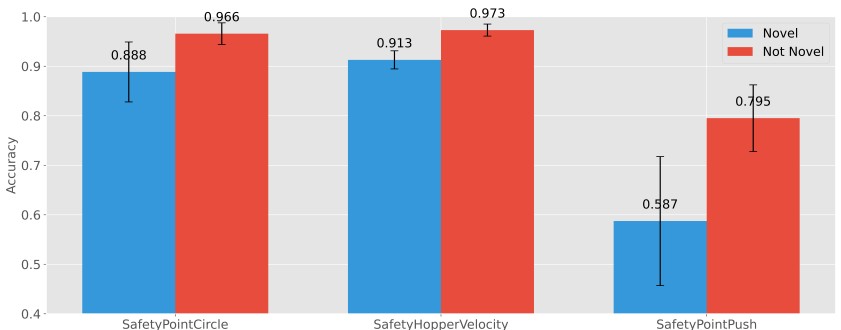

Figure 10: Model prediction accuracy (mean $\pm$ standard error) in the surrogate task (averaged over 3 seeds) with trajectory level feedback. Predictions are made on the next 50k steps following 250k training steps.

evaluated the model's accuracy on the pseudo-prediction task for these two trajectory types, as shown in Figure 10. This analysis indicates that the model's accuracy is lower for *novel* trajectories, corroborating the claims made in Section 4.3.

## D.4 Number of Queries generated

The number of trajectories queried for feedback is of the order of $\mathcal{O}(1e^3)$, the exact number differs across environments and individual runs due to novelty based sampling. Detailed information on the number of queries can be found in Table 4. We provide the two baseline methods with an advantage, whereby feedback is obtained for every trajectory generated by the policy, which is of the order $\mathcal{O}(1e^4)$.

## D.5 Complexity of Constraint Inference

Figure 11 highlights the difficulty of inferring constraints when cost violations are sparse. The brief interaction periods with obstacles in the environment make it challenging to assign credit to unsafe states when feedback is collected over long horizons. Consequently, feedback was collected at the state level in such settings. This is illustrated in Figure 12, where the overestimation bias is considerably

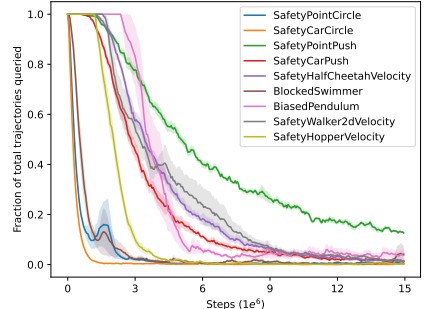

Figure 6: Implicit decreasing schedule observed when following novelty based sampling across different environments. The results are averaged over 6 independent seeds.

higher in the *Point Goal* environment compared to *Point Circle*, resulting in a significant drop in performance.

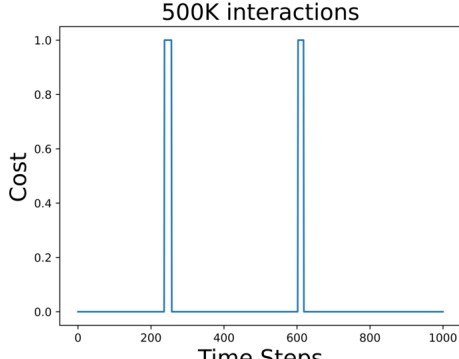 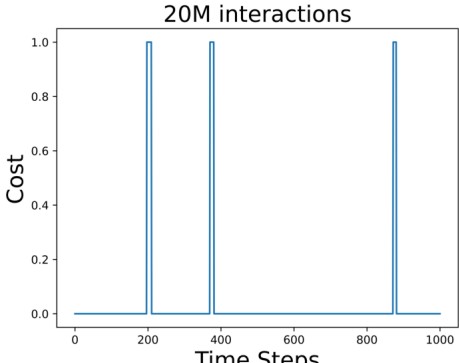

Figure 11: Costs incurred by a policy after 500K and 20M training interactions over a randomly sampled trajectory in the Point Goal environment.

## D.6    Ablation on the size of the feedback buffer

We conducted an ablation study on the size of the feedback buffer, the results of which are presented in Figure 13. We can observe that the policy becomes overly conservative when the size of the feedback buffer is small (100k, which corresponds to approximately two data collection rounds). We hypothesize that this is because of two interacting factors: (a) discarding past feedback can result in forgetting, which in turn leads to inaccuracies in estimating unsafe regions, and (b) there exists a feedback loop in which the inferred cost function influences on-policy data collection. If the inferred cost function overestimates costs in certain regions, the policy will avoid them and fail to gather the necessary feedback to correct this error. Conversely, if the costs are underestimated, the policy will visit those states and receive feedback, allowing for corrections. Therefore, smaller feedback buffer sizes can result in the learning of overly conservative policies.

## D.7    Need for the surrogate loss

In Section 4.2, we highlighted the challenges associated with minimizing Eq 3 particularly due to the term $\prod_{t=i}^{j} p^{safe}(s_t, a_t)$ which tends to collapse to 0 with longer segment lengths, resulting in unstable gradients. To investigate this further, we conducted a simple experiment in which we performed a few gradient steps to optimize Eq 3 and plotted the gradient norm. The results, shown in Figure 14 reveal that (a) directly optimizing Eq 3 produces gradients with a large norm that either quickly vanish or explode, rendering the learning process infeasible. In contrast, the surrogate objective in Eq 4 yields stable gradients allowing for convergence.

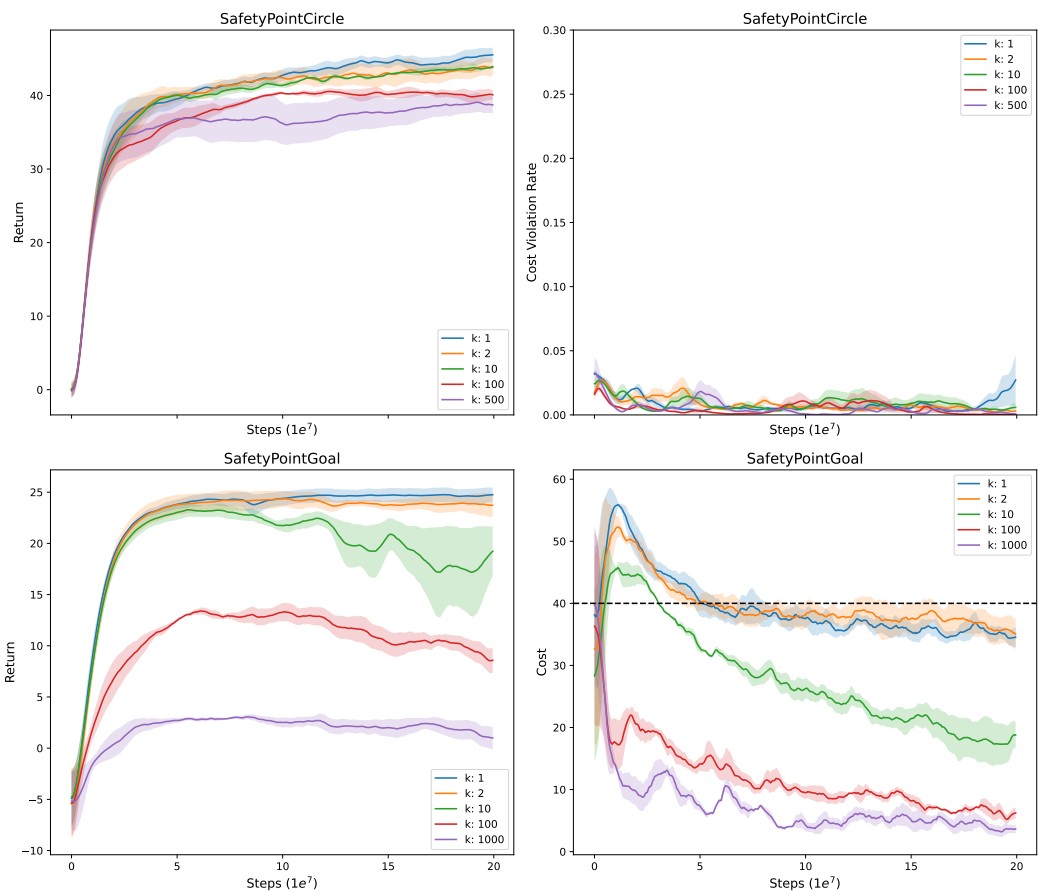

Figure 12: Ablation on the segment length for feedback elicitation.

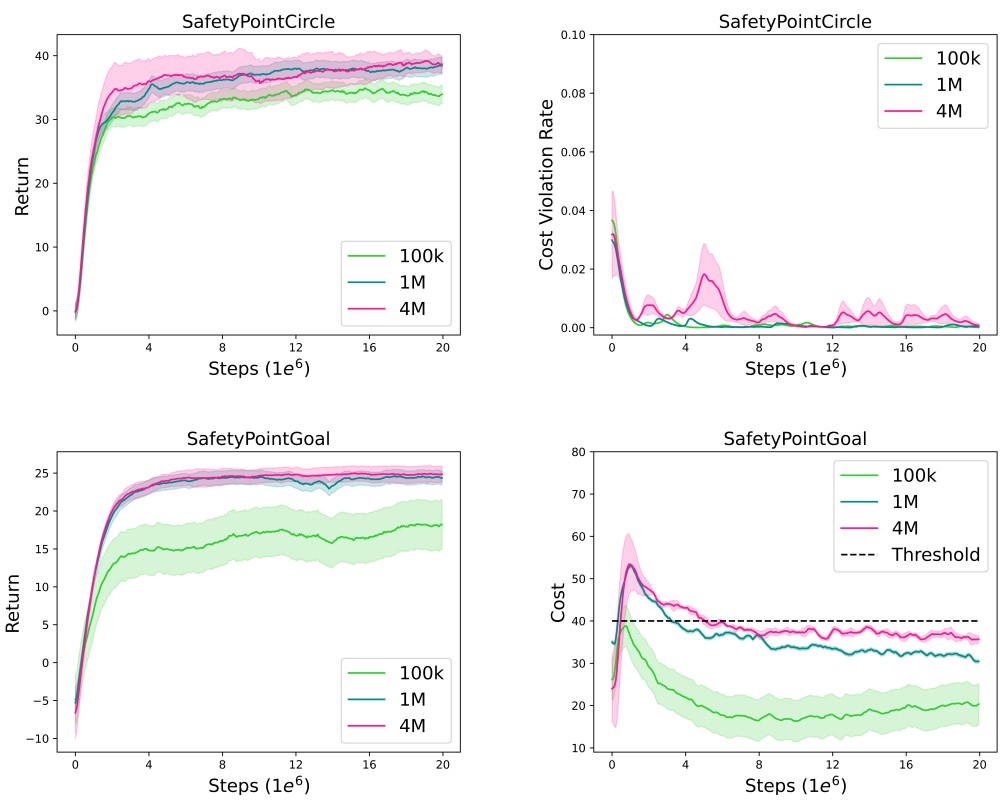

Figure 13: Impact of the size of the feedback buffer on performance.

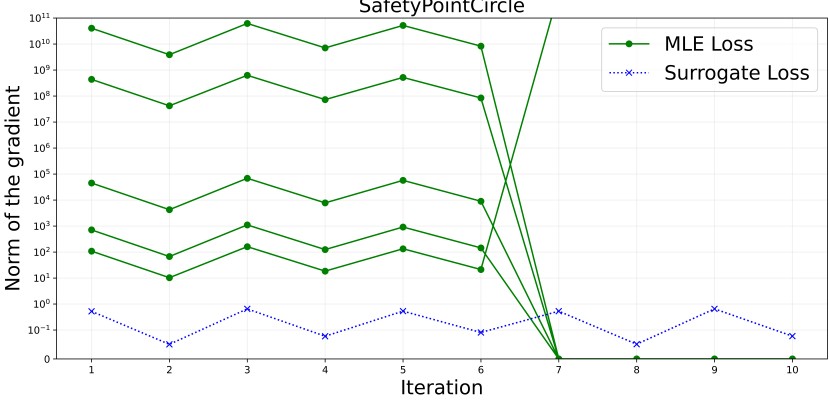

Figure 14: Norm of the gradient when optimising the MLE loss v/s the proposed surrogate loss with trajectory level feedback.

