# OpenReview forum: "Safety through feedback in Constrained RL"
_NeurIPS.cc/2024/Conference — NeurIPS 2024 poster_

### Official Review · Reviewer_xFMh · 2024-07-09

**Soundness:** 2
**Presentation:** 3
**Contribution:** 2
**Rating:** 4
**Confidence:** 3

**Summary:**

This paper studies the safe RL problem with an unknown cost function. As an on-policy approach, this work tries to learn the cost function with safety feedback from an evaluator with novelty sampling and conducts policy optimization at the same time. The authors propose that their approach can deal with feedback over longer horizons by a surrogate loss, reduce sampling queries by novelty sampling, and demonstrate the effectiveness via a set of experiments.

**Strengths:**

1. The paper is well-written and easy to follow in general.
2. The experiment results look promising.
3. The setting of the unknown cost function in safe RL is very interesting and important.

**Weaknesses:**

1. The novelty of this paper might not be enough. First, it is not clear to me the difficulty of inferring the cost function in RL, current existing approaches, and how the proposed approach outperforms the base approaches. Second, novelty sampling or frequency-based sampling is well-studied and leveraged in classical RL. Third, there is nothing new in policy optimization in this paper.
2. It is not clear to me that how close the learned cost function is to the ground truth cost function.

**Questions:**

1. In section 4.2, why do you formulate the safety/cost function by a probability measure? It is confusing to me. Therefore, I cannot understand why using maximum likelihood to learn the cost function.
2. You have to show the learned cost function is close to the ground truth in the experiments. Otherwise, I cannot trust the experimental results as there are so many uncertainties/hyper-parameters in RL.
3. A suggestion for this paper is to consider LLM as an evaluator.

**Limitations:**

The authors discussed their limitations in the conclusion.

---

> ### Author Rebuttal · Authors · 2024-08-07
>
> Thank you for your thoughtful review and constructive feedback. Your insights and suggestions are appreciated, and we look forward to addressing your questions below:
>
> 1) In binary classification problems, the ground truth is often represented as a Dirac measure (indicator function), and maximum likelihood estimation (MLE) leads to the widely used Binary Cross-Entropy (BCE) loss. Motivated by this, we represent the cost function as a probability measure in our work and subsequently use MLE.
>
> 2) We understand the concern and have added plots showing the inferred cost function and the estimated costs in Figure 1 in the attached document. It can be seen that the cost function closely approximates the true cost in all the environments. In Hopper, there is a bit of divergence, but with enough steps, we are able to get close.
>
>
> 3) Yes, this suggestion makes for interesting future work. The challenge lies in converting the MDP to text, instead eliciting feedback from multi-modal foundation models that are shown videos of the trajectory is another promising direction.
>
> 4) We understand this is a subjective assessment of the reviewer and we respect that. However, we do wish to point out that there are multiple key differentiating factors between the proposed work and existing literature that the reviewer seems to have missed:
>     * [25,29,10,7,5] are restricted to state-level feedback from the evaluator. Which can be a burden on the evaluator. The proposed method does not have the same restriction. Our results on the Circle and Mujoco based environments show that it is possible to collect feedback at the trajectory level and still obtain good performance.
>     * All existing methods except [7] do not consider the problem of selecting a subset of all the trajectories taken for feedback. Eliciting feedback for every trajectory taken can again lead to a significant burden on the evaluator.
> They do not scale to complex environments with large state-action spaces. To the best of our knowledge, our paper is the only one to experiment on complex environments like the Safety Gymnasium benchmark, which is used to evaluate the performance of Constrained RL algorithms with known cost functions.
>     * Scalable methods have been proposed in Constraint Inference from demonstrations [6,30,20]. These methods assume the existence of constraint-abiding expert demonstrations which may not be readily accessible in all scenarios. Instead of generating expert trajectories, we only need evaluation of automatically generated trajectories by human/machine.
>     * Novelty-based sampling has been introduced in the context of exploration problems [8,26]. To the best of our knowledge, we are the first to apply it in feedback-based RL settings. Additionally, we extend the concept of novelty to trajectories, inspired by its similarity to edit distance between trajectories [24]. The effectiveness of novelty based sampling as a form of sampling states that have a high prediction error is shown in Figure 5 in the attached document, and the overall gain in performance from using this sampling method is present in Figures 1 and 7.
>
>     We would refer the reviewer to Section D in the Appendix for further details on the prior work.
>
>     Our analysis of Table 1 reveals that the proposed RLSF algorithm demonstrates notable improvements over the baseline approaches across all tested environments. The baseline methods faced challenges in balancing cost estimation and reward optimization. In some cases (Point Circle, Biased Pendulum, Blocked Swimmer, Half Cheetah, Point Goal, Point Push), the baseline approaches  tended to underestimate costs, resulting in higher cost violations. In others (Car Circle, Car Push), costs were overestimated, leading to suboptimal reward outcomes.
>     The RLSF algorithm, however, showcases consistent superior performance compared to these baseline methods. Additionally, it achieves results comparable to PPOLag in several environments which underscores the potential of RLSF as a promising approach for addressing the complexities of reinforcement learning tasks without knowing the underlying cost constraints and just having demonstrations.

---

> > ### Comment · Reviewer_xFMh · 2024-08-10
> >
> > I appreciate the feedback from the authors. BCE and MLE are fine to the reviewer because they are common and standard for a binary classification task.
> >
> > Unfortunately, I don't see the current Figure 1 in the paper shows any comparison between learned cost function and ground truth, which further adds to my confusion.
> >
> > As for the multiple differentiating factors, I don't feel they are significant. Therefore, I will keep my current score.

---

> > > ### Author Response · Authors · 2024-08-11
> > > **Please Refer Figure 1 in the attached PDF not main paper.**
> > >
> > > It appears there's been a misunderstanding regarding Figure 1. We're specifically referring to Figure 1 in the PDF attached to the common rebuttal, not the Figure 1 in the paper. This figure shows that the inferred cost function is indeed close to the ground truth cost function.
> > >
> > > Regarding the differentiating factors, we respectfully disagree with the reviewer's assessment that they are insignificant. The challenge of designing cost or reward functions for every state-action pair is substantial, as there can be unintended consequences due to interaction of the costs. Our contributions are significant because they enable the integration of qualitative feedback on safety of trajectory segments, which is otherwise difficult to quantify. For example, consider a Roomba robot cleaning a house. If the owner indicates that a certain movement around the sofa was unsafe because the robot unexpectedly emerged from behind it, our approach can incorporate this feedback to prevent similar behavior in the future. This capability is crucial in creating systems that can adapt and improve based on real-world, qualitative safety feedback.

---

### Official Review · Reviewer_1oHB · 2024-07-17

**Soundness:** 4
**Presentation:** 3
**Contribution:** 3
**Rating:** 7
**Confidence:** 4

**Summary:**

This paper presents a method for using evaluator feedback as safety constraints in a constrained RL framework. The novelty comes from the nature of the feedback and the sampling scheme. It is motivated by the fact that previous approaches have various limitations: designing cost, especially comprehensively, is expensive and even seemingly impossible a priori. Offline feedback can help, but existing approaches often don't scale, and/or are constrained to receive feedback at the state level, which is also expensive. Giving feedback at the trajectory level is an option, but also a difficult state-credit assignment problem; and even getting feedback for every trajectory is expensive.

This paper presents RLSF, a method in which binary feedback is given on segments or entire trajectories and a novel surrogate loss is presented to make this a useful learning signal, then a new "novelty-based sampling mechanism" is used to train efficiently.

The paper takes us through the methods, including the feedback collection process (human or procedural evaluators), the policy improvement step, the nature of the feedback (1 if every state in a given segment is safe, 0 otherwise), and how the cost function is inferred from the collected data. It also presents a surrogate loss function for turning segment-level feedback into a dense signal.

The paper then goes through experiments and results in the Safety Gym and Driver simulation environments. The tasks have various constraints in position and velocity. Experiments compare to three general baselines - self-imitation safe RL, safe distribution matching, and PPO where the true cost function is known (considered an upper bound). The methods are tested directly on the tasks, then learned cost functions are transferred to a new agent. The paper then goes through ablations removing the novelty-based sampling method. The results show that this RLSF algorithm shows significant improvement over baselines.

**Strengths:**

### Originality
- I am not aware of any methods that use exactly this surrogate loss or sampling method. RLSF does borrow heavily from prior work, but the loss function, sampling method, and otherwise novel (to my knowledge) mix of existing elements does constitute an original work.

### Quality
- Experiments section, particularly ablation of the sampling section and cost transfer, is creative and shows lots of important aspects of the study. The cost transfer results are particularly promising.
- Results speak for themselves! They show a system that works in simpler environmetns and is ready to be scaled up and tested.

### Clarity
- Well written and clear. There is a fair amount of technical content in terms of defining problems and setting up notation, and the paper handles it all well.
- The novelty-based sampling mechanism is particularly well-explained.

### Significance
Seems good. It's good to see new ways of integrating feedback as human interaction with robots comes nearer and nearer.

**Weaknesses:**

### Quality
- "We posit that with a sufficient number of samples, the number of correct labels will outweigh the number of incorrect labels for each state" - doesn't this require sparsity of safety violations and small segments (meaning a hefty feedback load)? Questions like this would require answers to judge if this work is useful.
- It would be good to see more error modes - for example, the paper cites novel states as a hotbed for potential errors, but it's only one. What are the others?
- In the experiments section, it would be good to see cost transfer compared to other cost function learning methods, not just PPO vs. RLSF vs. PPOLag.

### Clarity
- Tables should have bolded numbers, even if all the bolding will be in one column. The tables are incredibly dense right now and tiring to read.
- Explanation of surrogate loss derivation would benefit from more justification. E.g. I see that it's true that the product term substitution makes it a state-level term instead of a segment-level term, but explain that. more clearly and why it's okay to do and still expect the optimization to work out.

**Questions:**

- The paper cites sparse cost violation as a motivation for collecting feedback on shorter segments. Why is that? Is the idea that when cost violations are dense, it's hard to give more specialized feedback, but when they're more sparse then the evaluator can just go for it? Or is it purely a horizon issue (shorter horizons have fewer cost violations), in which case mentioning sparse cost violations at all doesn't make sense?

---

> ### Author Rebuttal · Authors · 2024-08-07
>
> Thank you for your thorough and positive review of our paper. We look forward to addressing your questions and suggestions below:
>
> 1) By sparsity we mean the density of violations within the segment is low. This makes distinguishing safe states from unsafe states challenging, as it becomes difficult to identify which states caused the segment to be marked unsafe. Consider Figure 8 in the Appendix, in this case less than 10% of the states in that trajectory are actually unsafe. Thus, the rest of the states get an incorrect pseudo-label of unsafe. This would reduce significantly if the cost violations were more dense, say (~50%) of the states within the trajectory were unsafe, or if feedback was collected over shorter segments (which inherently reduces sparsity). To summarise, dense cost violations and shorter feedback segments simplify the cost inference problem, while sparse violations and longer segments make it more challenging.
>
>
> 2) Thanks to the reviewer for pointing this out and we do need to modify it as follows:
>
>     “We posit that with a sufficient number of samples, we will be able to distinguish the safe states from unsafe states. This is because we observe that:
>
>     (i) safe states will appear in both safe and unsafe segments; and
>     (ii) unsafe states would appear only in unsafe segments and not in safe segments;”
>
>     Hence with a sufficient number of samples,  it would become feasible to distinguish safe from the unsafe states.
>     We would direct the reviewer to our strong performance on multiple safety environments (Circle, Goal and Mujoco based) that                 indicates this hypothesis to hold quite well in environments when the feedback is collected for the entire trajectory.
>
> 3) We can broadly categorise the reasons for model prediction errors into four main branches: Limited Data, Noise in the Labels, Model Misspecification and Optimization Error [Burda, et al.]. The latter two sources must be addressed before the learning process. Novelty based sampling addresses the first source of error. Label Noise can be reduced by decreasing the segment length. Other methods such as eliciting extra state-level feedback on states with noisy labels are also possible, making for interesting future work.
>
> 4) Please note that for doing cost transfer, there is an explicit cost function that has to be learnt that can be transferred. We would like to highlight that SIM and SDM do not learn explicit cost functions. Additionally, previous work [25,29,10,7,5] has not been scaled to handle the environments with continuous state and action spaces considered in these experiments. Therefore, our cost transfer experiments (Table 2) were limited to the aforementioned algorithms.
>
> 5) We appreciate the feedback on Table 1, we will bold the numbers in the final draft.
>
> 6) The main motivation for proposing the surrogate loss was due to the numerical instability that arises from multiplying the probabilities over a long horizon. To further illustrate this point, we plotted the norm of the gradient after the first 10 epochs of training the classifier using Eq 3 directly, as shown in Figure 1 (in the attached document). The gradient either collapses to 0 or explodes very quickly in the first few iterations making optimization challenging. The surrogate objective side-steps the need to multiply probabilities over long horizons and thus results in stable gradients as shown in Figure, leading to easier optimization. We acknowledge that this motivation was brief in the paper (due to space limitations), but will take this feedback into consideration and make the motivation more clear.
>
> Citations:
> Burda, Y., Edwards, H., Storkey, A., & Klimov, O. (2019). Exploration by random network distillation. In International Conference on Learning Representations (ICLR).

---

> > ### Comment · Reviewer_1oHB · 2024-08-13
> > **Thanks for the response**
> >
> > Hi, thanks for the detailed response. All the changes look good. They were mainly about presentation and detail, so they wouldn't move me to change my score, but I do think they make the paper a better read.

---

### Official Review · Reviewer_w6J6 · 2024-07-30

**Soundness:** 2
**Presentation:** 2
**Contribution:** 2
**Rating:** 5
**Confidence:** 2

**Summary:**

This paper proposes a surrogate loss function, which instead of collecting a feedback over trajectory-based evaluator, it breaks long trajectories into segments and the evaluator classifies and assigns labels to segments unsafe if any individual state is unsafe during the segments. This paper also proposes a novelty sampling, which only gathers novel trajectories for feedback from evaluator based on the trajectory distance measure edit distance.

**Strengths:**

The pseudo algorithm helps to understand the paper's idea a bit more. The experiment part seems complete with the methods tested in different tasks and environments to cover the proposed algorithms and the novelty-based sampling method.

**Weaknesses:**

1. I found it hard to follow the paper, especially in section 4.2. Some questions are raised in the following section.
2. I understand that Eq 4 is an upper bound for Eq3, but I doubt that the difference between Eq3 and Eq4 might be too huge, and in this case, switching to Eq4 might result in a very inaccurate estimate $p_*^{\mathrm{safe}}$ for the following analysis in the paper.

**Questions:**

1. On line 136, the probability of a state being safe is defined as $\mathbb{1}[c_{\mathrm{gt}}(s, a) = 0]$. Isn't this an indicator function instead of assigning a probability to this specific state? Or this is a typo.
2. What does the ``noisy labels`` mean on line 150? This phrase also appears somewhere else and I am not sure what it refers to. Also for the sentence following that, ``However, if it occurs in a segment labeled unsafe, it is incorrectly labelled unsafe``. Does it mean if the segment is labelled as unsafe, some of the individual states in the segment ought to be safe but they are labeled as unsafe then?
3. Why is it true that ``with a sufficient number of samples, the number of correct labels will outweigh the number of incorrect lables for each state``?
4. I can't quite follow the paper but based on my understanding, there's a human evaluator to evaluate segment, and another evaluator for state?

**Limitations:**

Both limitations and societal impacts are discussed in the paper.

---

> ### Author Rebuttal · Authors · 2024-08-07
>
> Thank you for your detailed review and constructive feedback. We appreciate your insights and will address your questions and concerns below:
>
> 1) Please note that $p_{gt}^{safe}(s,a)$ is the ground truth probability. Since ground truth is exactly known for a state on whether it is safe or unsafe, $p_{gt}^{safe}(s,a)$ is either 0 or 1 depending on whether $c_{gt} = 0 \text{ or } 1$ respectively. This is the reason for using indicator function for ground truth.
> $p^{safe}(s,a)$ is our estimate of the probability and this will typically have values that are not 0 and 1 only.
>
>
> 2) The surrogate loss introduces a binary classification problem where each state is assigned a pseudo-label $y^{\text{safe}}$ based on the segment it occurred in. Safe states (w.r.t ground truth cost) can be assigned both unsafe and safe pseudo-labels depending on whether they occur in a segment that contains an unsafe state (or not). We refer to this phenomenon as “noisy labels” in the surrogate classification task.
> Yes, the evaluator is tasked to classify a segment unsafe if it contains one or more unsafe states. The rest of the states (if any), although inherently safe, are assigned the pseudo-label of unsafe.
>
>
> 3) Thanks to the reviewer for pointing this out and we do need to modify it as follows:
> “We posit that with a sufficient number of samples, we will be able to distinguish the safe states from unsafe states. This is because we observe that:
> (i) safe states will appear in both safe and unsafe segments; and
> (ii) unsafe states would appear only in unsafe segments and not in safe segments;”
> Hence with a sufficient number of samples,  it would become feasible to distinguish safe from the unsafe states.
> We would direct the reviewer to our strong performance on multiple safety environments (Circle, Goal and Mujoco based) that indicates this hypothesis to hold quite well in environments when the feedback is collected for the entire trajectory.
>
>
> 4) There is a single evaluator (human or a program that is pre-trained/pre-coded) that classifies a segment. Each trajectory that requires evaluation (chosen based on novelty) is then broken down into segments, and each segment is passed to the evaluator for feedback. Feedback is not explicitly collected for individual states unless the segment length is 1.
>
>
> 5) We theoretically quantify the difference caused by transitioning from Equation 3 to Equation 4 in Proposition 3, in terms of cost overestimation. If this difference is considered excessive (results in overly conservative policies), the segment length can be reduced (by the user). However, this adjustment comes at the cost of increased evaluation effort, thus introducing a tradeoff.
>
>     Directly optimising Equation 3 with trajectory-level feedback results in numerical instability (Figure 2, attached document). This highlights the necessity of the proposed surrogate objective.
>
>     We further investigate the effects of transitioning from Equation 3 to Equation 4 in Table 1 of the attached document. MLE corresponds to taking feedback at the state level, equivalent to solving Equation 3. The Surrogate, on the other hand, represents solving Equation 4 with feedback collected at the trajectory level.
>
>     We find that in the Circle environment, optimising the surrogate loss with trajectory-level feedback achieved performance comparable to optimising Equation 3. This suggests that the difference between Equation 3 and Equation 4 is minimal in the Circle environment. Additionally, the strong performance of RLSF compared to PPOLag in Table 1 of the main paper indicates that this minimal difference likely extends to the other Mujoco-based environments as well, demonstrating the promising applicability of our approach across a wide range of problems.
>
>     However, Table 1 in the attached document shows a divergence between the two objectives in the Goal environment. The poor performance of RLSF in this case can be attributed to the overestimation bias discussed in the paper and the complexity of the cost function in this task, as described in lines 267-275. Therefore, in this scenario, state-level feedback is necessary for good performance.

---

> > ### Comment · Reviewer_w6J6 · 2024-08-13
> >
> > I thank authors for addressing my questions. I am not an expert in this field, and I'm willing to raise my score from 4 to 5.

---

### Official Review · Reviewer_E6xa · 2024-07-31

**Soundness:** 3
**Presentation:** 3
**Contribution:** 3
**Rating:** 7
**Confidence:** 4

**Summary:**

Authors propose a new way of estimating the cost function for constrained RL through user feedback. They propose a surrogate loss that is used to train a model that estimates the probability that a state-action pair is safe. They then use the model to estimate the cost of a policy and adhere to the constraint while looking for the best reward.

**Strengths:**

The paper is relatively easy to follow for a person that is not very familiar with the field. The motive and theoretical assumptions are explained fairly well.

**Weaknesses:**

The experiment section lacks some details and is much harder to follow.

I did not see any experiments (ablations) on the size of the feedback buffer.

Also I was hoping to see some discussion around the introduced bias in estimation with differing c_max values.

**Questions:**

Could authors explain the DNN that estimates the probability of safety for a (s, a) pair?
seems like they are directly estimating the probability?
I am curious, if the authors have considered estimating the d_g and d_b densities (as defined in proposition 2 for the optimal solution) instead?

**Limitations:**

How is novelty sampling (which seems naive to me) related to uncertainty sampling? The proposed novelty sampling, roughly, ignores the trajectories that have many visited sub-sequences.

I think Corollary 1. is theoretically wrong, as the authors have conveniently ignored the inherent inaccuracies in user feedback. Authors mention labeling noise as the noise that is the result of expanding an unsafe label to all the pairs in the trajectory but there is also the meta noise that the user labeling(user feedback) by itself is an error-prone process.

---

> ### Author Rebuttal · Authors · 2024-08-07
>
> Thank you for your detailed and constructive review of our paper. We appreciate your positive assessment of our work and the valuable feedback you provided. We hope to address the weaknesses and questions you raised below:
>
> 1) We use a regular Multi-Layer Perceptron (MLP) to estimate the probability of safety. The model takes the state-action pair as input and outputs the probability $p^{safe}_{\theta}(s,a)$. We request the reviewer to refer to Section C.1 in the Appendix for further details on the network architecture of the Classifier.
>
>     Yes, we did consider directly estimating $d_{g}$ and $d_{b}$, some possible methods include:
>     * SimHash can be used to discretize the state-space and store $d_{g}$ and $d_{b}$ via counts. We believe hashing states via raw feature state features is good enough for novelty estimation but the classification task would require more rich features. This is because this type of model is essentially a linear model, limiting its capacity to capture state densities at the accuracy required for cost function estimation.
>     * Exemplar methods, as proposed by Fu et al., estimate state densities through a proxy binary classification task. This task distinguishes between a single state $s^*$ and the states in the feedback buffer $B$. Let $D$ represent the dataset containing only $s^*$. Since $s^*$ may also exist in $B$, $p(D|s^*)$ is not necessarily 1, requiring the classifier to account for the density of $s^*$ in $B$. This approach enables the estimation of $p(s^*|B)$, i.e, the state density. We believe solving two proxy binary classification problems (one for each buffer) is inefficient compared to the single problem specified by Eq 4.
>     * Generative Models such as GANs [Goodfellow, et al.], VAEs [Kingma, et al.] and Normalising Flows [Rezende, et al.], etc. that implicitly store the densities in order to generate samples. The density can be extracted from Flow based models but GANs and VAEs require additional modifications to extract the density. These models are more complex and hence introduce additional constraints such as increased data requirements and computational demands.
>
>     Given the above reasons, we believe directly estimating the probability $p^{safe}_{*}$ is a more suitable approach.
>
> 2) Thanks for pointing this out. We have added an ablation on the size of the feedback buffer in the attached document (Figure 3). As expected, we notice that the performance of RLSF increases with the size of the feedback buffer as feedback given in the past is not discarded.
>
> 3) We agree that a discussion on this point would be insightful and will add it in the revised version. Corollary 1 is true for any threshold $c_{\text{max}}$, since the estimated cost function has an overestimation bias. Consequently, satisfying the $c_{\text{max}}$ threshold can lead to overly conservative policies. To counter this, ideally one could increase the threshold $c_{\text{max}}$ to $c_{\text{max}} + \text{bias}$. Calculating the bias apriori without the knowledge of the ground truth cost function is infeasible. Hence, one could take a heuristic approach and increase the threshold to $c_{max} + \delta$, with $\delta \in \mathbb{R}$. We conducted additional experiments in the SafetyCarCircle environment to test this method (Figure 4 in the attached document), as we observed higher overestimation bias in this setting (Table 1 in the main paper). We observed that adding such a bonus does indeed boost the performance of the proposed method.
>
> 4) Novelty-based sampling is a form of uncertainty sampling as it targets states with high epistemic uncertainty—uncertainty arising from a lack of feedback. Epistemic uncertainty is known to adversely affect prediction accuracy [Burda, et al.]. Since we cannot directly measure epistemic uncertainty, we instead analyse the correlation between novelty of trajectories and the model's prediction accuracy on states within those trajectories. This analysis, presented in Figure 5 of the attached document, serves to evaluate the effectiveness of our proposed novelty-based sampling in identifying states with high epistemic uncertainty.
> The results demonstrate that our novelty-based sampling method does indeed identify states with high predictive errors, which is indicative of high epistemic uncertainty. By eliciting feedback for these states, we can improve the model's accuracy on these states. This further explains the superior performance of our proposed sampling method, as evidenced in Figures 1 and 7 of the main paper.
>
> 5) Please note that Corollary 1 is assuming ground truth, $c_{gt}$ information is correct (and not misspecified), the feedback is “sufficient” and is based on the optimal classifier $p^{safe}_{*}$
>
>     We acknowledge that real-world scenarios may deviate from these assumptions:
>      * $p^{safe}_{\theta}$ \text{ may not always converge to } $p^{safe}_{*}$ due to factors such as optimization errors, model capacity, etc.
>      * Feedback may be missing for some states.
>      * Evaluation (meta) noise may occur, particularly with human evaluators. Possible methods to account for this (meta) noise are:
>
>        a) $p^{safe}_{gt}(s,a)=\mathbb{I}[c_{gt}(s,a)=0]$ (noise in safety preferences)
>
>        b) In $y^{safe}$ (noise in labelling process)
>
>     We agree that theoretical results encompassing all these factors would be valuable. However, developing such a comprehensive analysis is non-trivial and beyond the scope of this work. Hence, we have also provided experimental results for cases where these assumptions cannot be satisfied.
>
> Citations: \
> Burda, Y., et al.. Exploration by random network distillation. ICLR. \
> Fu, J., et al. EX2: Exploration with exemplar models for deep reinforcement learning. NeurIPS \
> Goodfellow, I., et al.. Generative adversarial nets.  NeurIPS \
> Kingma, D. P., et al. . Auto-Encoding Variational Bayes. ICLR. \
> Rezende, D. J., et al. Variational Inference with Normalising Flows. ICML

---

### Author Rebuttal · Authors · 2024-08-07

We thank all the reviewers for their thoughtful and insightful reviews. We're happy to see that reviewers E6xa, 1oHB, and w6J6 found our paper well-written and easy to follow, and that all reviewers recognized the importance of our work in the context of safe reinforcement learning.

---

### Decision · Program_Chairs · 2024-09-25

**Decision:**

Accept (poster)

**Comment:**

This paper presents a novel approach to learning cost functions for constrained reinforcement learning from trajectory-level feedback, which is an important problem in safe RL. The proposed Reinforcement Learning from Safety Feedback (RLSF) algorithm shows strong performance across a range of benchmark environments, including Mujoco-based and Safety Gymnasium tasks. Based on the reviews and the authors' responses, I recommend accepting this paper though with some reservations.

The reviewers generally found merit in the paper's approach to learning cost functions for constrained reinforcement learning from trajectory-level feedback. The main strengths highlighted by the reviewers include the novel formulation of the problem, the ability to learn from coarse feedback, and the strong performance across a range of benchmark environments, including complex Mujoco-based tasks. The novelty-based sampling mechanism for efficient feedback collection was also viewed positively.

However, some limitations were noted across the reviews, such as about the novelty of the work and the lack of comparisons to other learning-based methods for cost inference. The authors have clarified several key differentiating factors of their work, including the ability to use trajectory-level feedback, selective feedback collection for scalability, and the novel application of novelty-based sampling in feedback-based RL settings. The authors' rebuttal provided some clarifications and additional results, including ablation studies on different data sources and visualizations of the learned cost functions. These additions help address some of the reviewers' concerns, particularly regarding the effectiveness of the method and its comparison to baselines.

To improve the final version, the authors should:

1. Include more comparisons to relevant cost inference methods.
2. Provide the additional ablation studies and visualizations mentioned in the rebuttal in the main paper.
3. Clarify the technical details that were found confusing, particularly around the surrogate loss derivation.
4. Address the limitations more thoroughly, especially regarding the simplicity of some experiments and the reliance on simulated feedback.

While the concerns about limited comparisons and experimental simplicity are valid, the overall positive assessment from the majority of reviewers and the potential impact of this work on safe RL justify acceptance. In conclusion, despite some limitations, this paper presents a novel and important contribution to the field of safe reinforcement learning, demonstrating a promising approach to learning cost functions from coarse feedback.